# FIRE: Learning to Navigate and Act on Real-World Files via Stateful Reinforcement Learning

**Jingyuan Ma**[1][*]  **Zheng Li**[1][*]  **Rui Li**[1][*]  **Siyao Song**[2]  **Ying Zeng**[2]  **Minghao Li**[2][†]  **Zhifang Sui**[1][†]

## Abstract

Large language models still struggle to reliably answer questions grounded in real-world files like spreadsheets and slides, where evidence is scattered across irregular layouts and heterogeneous formats. We address this by formalizing File Reasoning, a setting where agents must interact directly with unprocessed files (XLSX, PDF, DOCX, PPTX) within a persistent sandbox. To support this, we introduce a unified data pipeline and a high-difficulty benchmark of over 400 verifiable questions that preserve native file structure. Furthermore, we propose a reinforcement learning framework grounded in stateful file execution. We train **FIRE** (**F**ile **I**nteractive **R**easoning **E**xpert), a family of models that learn to optimize long-horizon planning using genuine execution feedback. Unlike stateless tool-use methods, this environmental persistence enables agents to iteratively recover from errors and adapt to structural ambiguities. Empirical results show that Qwen3-32B-FIRE achieves the strongest performance among the open-source models of comparable scale under identical execution constraints.

## 1. Introduction

The capabilities of large language models (LLMs) have expanded from simple dialogue to complex agentic workflows (Huang et al., 2024; Yao et al., 2023; Schmidgall et al., 2025). However, a critical gap remains in their ability to operate in file-centric environments, which constitute the backbone of real-world knowledge work. Unlike simplified text inputs, real-world files, such as multi-sheet spreadsheets, formatted reports, and slide decks, are not merely containers of text but structured, stateful environments. Solving tasks in this domain requires more than passive reading; it demands active file reasoning: navigating hierarchical layouts, executing dependencies (e.g., formulas), and synthesizing evidence scattered across heterogeneous formats.

Despite advances in tool use (Wei et al., 2022; Li et al., 2023), current models struggle to generalize to these settings. A primary reason is the reliance on the "parse-then-reason" paradigm, where documents are converted into flattened text or simplified representations (e.g., JSON/Markdown) prior to reasoning (Chen et al., 2020; Tang et al., 2025). While convenient, this approach suffers from two fundamental limitations. First, it artificially simplifies the task, effectively offloading the critical burden of structure parsing and information localization from the model to the preprocessing pipeline. By providing pre-digested content, these benchmarks fail to evaluate the model's intrinsic capability to navigate raw file environments. Second, such conversion is inherently lossy (Simmering et al., 2025). Flattening a multi-dimensional file, such as a spreadsheet with inter-cell dependencies or a slide deck with spatial layouts, inevitably strips away structural nuances and formatting cues that are often essential for deriving the correct answer. Consequently, when models face unprocessed files in the wild, the absence of these external aids leads to sharp performance degradation.

We argue that to build truly robust agents, we must shift from "reading files" to "acting with files." We formalize this through a File Reasoning Setting, where agents interact directly with raw XLSX, PDF, DOCX, and PPTX files within a persistent, executable sandbox. In this setting, the agent must autonomously explore the file structure, issue executable tool calls (e.g., Python scripts or Shell commands), and interpret execution feedback to verify its findings. To enable scalable research in this direction, we develop a unified pipeline that automates the collection, generation, and executable verification of file-based tasks. Using this pipeline, we construct a high-difficulty benchmark of over 400 tasks. Unlike existing datasets that rely on clean, pre-processed inputs (Mialon et al., 2024; Simmering et al., 2025), our benchmark preserves the native structural ambiguity of real-

---

[*]Work done during internship at ByteDance BandAI. [†]Corresponding authors. [1]State Key Laboratory of Multimedia Information Processing, School of Computer Science, Peking University, Beijing, China [2]ByteDance BandAI, Beijing, China. Correspondence to: Minghao Li <liminghao.bd@bytedance.com>, Zhifang Sui <szf@pku.edu.cn>.

*Proceedings of the 43rd International Conference on Machine Learning*, Seoul, South Korea. PMLR 306, 2026. Copyright 2026 by the author(s).

world documents, forcing agents to handle exceptions and messy formats.

Furthermore, we posit that supervised imitation is insufficient for mastering these interactive behaviors. File reasoning involves long-horizon planning where early actions (e.g., generating intermediate files or modifying document contents) permanently alter the environment state and determine the validity of future steps. To address this, we establish a reinforcement learning framework grounded in persistent, stateful execution. We train **FIRE** (**F**ile **I**nteractive **R**easoning **E**xpert), a family of models optimized via reinforcement learning directly within the sandbox. Empirical results show that Qwen3-32B-FIRE achieves state-of-the-art performance among the same scale open-source models, demonstrating the efficacy of our training framework.

Our contributions are as follows:

- We formalize File Reasoning as an interactive setting where agents operate on unprocessed files via sandboxed tools, moving beyond the limitations of static parsing.

- We introduce a scalable data pipeline and a challenging benchmark of over 400 verified tasks across four file formats, preserving native structural complexities.

- We propose a reinforcement learning framework based on persistent, stateful execution. We train **FIRE**, which demonstrates that learning from environment feedback significantly outperforms supervised baselines.

## 2. Related Work

**Tool Use by Language Models**    Recent advances have highlighted the growing ability of large language models (LLMs) to use external tools such as APIs, calculators, code interpreters, and search engines. Early prompting-based approaches, including ReAct (Yao et al., 2023) and related reasoning-then-acting methods (Wang et al., 2024; Yang et al., 2023), demonstrated that LLMs could learn to plan and invoke tools via language alone. More recent work has introduced structured tool interfaces using function calling or API schemas, improving robustness and compositionality. Fine-tuned models such as Gorilla (Patil et al., 2024), ToolLlama (Qin et al., 2024), and OpenAI's function-calling GPT-4 exemplify this shift toward more grounded tool use. Benchmarks like API-Bank (Li et al., 2023) and ToolBench (Xu et al., 2023) further support standardized evaluation of tool use across planning, selection, and execution. However, many existing studies on tool use focus on highly structured API calls with clearly defined parameters, such as filling in temperature or location for hotel search, which often reduce the agent's role to simple slot filling rather than genuine reasoning or decision-making.

**File Understanding and Manipulation by LLMs**    Most existing work on table and document reasoning, such as TabFact (Chen et al., 2020), Open-WikiTable (Kweon et al., 2023), WikiTableEdit (Li et al., 2024), TableLLM (Zhang et al., 2025), and SPIDER (Yu et al., 2018), relies on structured preprocessing to regularize complex files into clean formats like CSVs or text templates. In these settings, models perform reasoning over simplified representations rather than parsing raw documents, thus testing language understanding more than file comprehension. SSTQA (Tang et al., 2025) moves one step closer to real-world files with a hierarchical tree structure for semi-structured tables, but it still depends on explicit parsing and reconstruction. Similarly, GAIA (Mialon et al., 2024) and MeetBench (Simmering et al., 2025) extend to multi-document workflows while providing preprocessed text inputs. In contrast, our work goes beyond this "parse-then-reason" paradigm by adopting a fully end-to-end setting, allowing the model to directly interpret and act on unprocessed files.

## 3. Unified Framework for File-based Task Generation

### 3.1. Motivation and Overview

Constructing verifiable reasoning tasks from real-world files presents a dilemma: manual annotation is difficult to scale, while template-based generation often simplifies the inherent structural noise of documents. To overcome this, we build a scalable pipeline (Figure 1) guided by a core principle: preserving native file structure while ensuring ground-truth validity. Our framework treats raw files (e.g., heavily formatted spreadsheets, slide decks) as the primary environment, ensuring that agents are trained and evaluated on the procedural complexity of real-world workflows rather than sanitized text representations.

### 3.2. Question Generation via Asymmetric Views

We employ an asymmetric generation strategy to align high-quality logic with challenging file formats.

- **Tabular Files (XLSX):** We source data that exists in dual formats: a structured representation (e.g., CSV/HTML) and its corresponding native XLSX file. During generation, the teacher model is given the structured view to produce unambiguous question-answer pairs. However, at inference time, the agent is presented exclusively with the raw XLSX file. This ensures that the ground truth is mathematically precise, while the agent is forced to handle the messy reality of multi-sheet navigation and formatting irregularities.

- **Document Files (PDF/DOCX/PPTX):** For document-style files, we use a Vision-Language Model (VLM)

*Figure 1.* Overview of our proposed framework. The pipeline begins with the collection of diverse file formats (e.g., PDF, CSV). Subsequently, LLMs are utilized to generate questions, which undergo a verification process to ensure they meet quality requirements. Finally, the agent acts within a sandbox environment to interact with real files, undergoing training through a Cold-start (SFT) phase for tool learning and a Stateful Reinforcement Learning algorithm to optimize performance.

during question generation to inspect visual layouts and identify information-rich regions in the source documents. This helps the teacher model design questions that reflect realistic document structure, such as section hierarchy, tables, captions, and slide organization. At the same time, we explicitly instruct the generator and verifier to avoid questions that require purely visual recognition, such as estimating values from images, interpreting diagrams without textual support, or recognizing image-only content. This is because our current goal is primarily intended to assess file-grounded reasoning and tool-based information extraction, rather than general multimodal perception capabilities.

### 3.3. Executable Verification

To filter out hallucinations and ambiguous queries, we implement a rigorous Executable Verification stage. Mere linguistic checking is insufficient for reasoning tasks. Instead, we deploy a secondary "Auditor Agent" equipped with code execution capabilities. For every generated (Question, File, Answer) triplet, the Auditor attempts to solve the task by writing and executing code against the source data. We retain a task only if: (1) The Auditor successfully executes the code without errors; (2) The Auditor's derived answer matches the generated ground truth; (3) The answer is unique and deterministic. This execution-based filtering guarantees that every task in our benchmark is fundamentally solvable via programmatic reasoning.

### 3.4. Continuous Data Expansion

The framework is connected to large-scale web sources (e.g., Common Crawl) to support autonomous data expansion. Newly ingested files are processed through this generation-verification loop automatically. This design allows our training corpus to grow organically, exposing the model to an ever-expanding diversity of real-world file headers, encodings, and layouts, thereby preventing overfitting to specific document templates.

### 3.5. Benchmark Construction

Building upon the unified generation framework, we curate a benchmark designed to rigorously evaluate LLMs in real-file environments.

**Sources and Coverage**   We curate a diverse corpus of real-world files spanning four formats: **XLSX, PDF, DOCX, and PPTX**. Sources include authoritative government portals (e.g., U.S. Census Bureau), financial filings (CNINFO), and open-access repositories (WikiTables, Zenodo). Unlike datasets that rely on synthetic templates, these files retain native irregularities such as merged cells, inconsistent headers, and complex vector layouts.

**Filtering Strategy.**   To ensure the benchmark is both discriminative and correct, we apply a rigorous filtering pipeline to the generated candidate pairs:

*Table 1.* Dataset statistics by file type. Pg. denotes pages or slides; Sz. denotes file size in megabytes; #Qs denotes the number of questions. XLSX files have no page counts so page statistics are omitted.

| Type | #Files | Mean Pg. | Max Pg. | Avg Sz. (MB) | #Qs |
|------|--------|----------|---------|--------------|-----|
| PDF | 74 | 15.91 | 36 | 0.84 | 87 |
| DOCX | 36 | 4.58 | 19 | 0.13 | 67 |
| PPTX | 45 | 22.42 | 55 | 6.91 | 60 |
| XLSX | 134 | – | – | 0.24 | 196 |

- **Consensus & Adversarial Filtering:** We attempt to solve each task using three strong models (GPT-5, DeepSeek-V3.1, Gemini-2.5-Pro) with a pass@1 setting. We discard trivial tasks (solved by all models) and ambiguous tasks (where models produce conflicting "correct" answers). We retain tasks that are solvable by at least one model but challenging for the majority.

- **Manual Auditing:** For the retained set, human experts manually inspect file integrity (e.g., absence of corrupted encodings), verify logical consistency, and perform necessary rewriting or correction to ensure clarity, correctness, and coherence.

We exclude files that require image perception or are excessively long (hundreds of pages), focusing the evaluation on text-grounded structure and executable reasoning within a manageable scope.

**Final Statistics**  After multi-round filtering and corrections, the benchmark contains 410 verified questions grounded in valid, interpretable files. Table 1 summarizes distribution and statistics across formats, showing that the benchmark covers a wide spectrum of real-world file types and scales, from short textual documents to multi-sheet spreadsheets and multi-page presentations. This diversity ensures balanced coverage and structural variability, making the benchmark both realistic and challenging for file reasoning.

## 4. Training Framework

### 4.1. Overview

We adopt a two-stage training framework consisting of supervised fine-tuning (SFT) followed by reinforcement learning (RL). This design reflects a natural progression from learning how to use tools to learning how to use them effectively in real file-based reasoning tasks.

In the SFT stage, the model is trained on example trajectories that demonstrate basic tool usage and structured interaction patterns. These trajectories teach the model when to invoke tools, how to format tool calls, and how to interpret execution results, providing the necessary foundation for file-based reasoning. In the RL stage, the model further improves its decision-making and planning abilities through trial and error in a sandboxed file environment. This stage encourages exploration and allows the model to refine its reasoning strategies beyond supervised demonstrations.

Together, this two-stage scheme enables a transition from imitation to autonomous problem solving, which is essential for complex tasks involving real-world files.

### 4.2. Cold-start Supervised Fine-tuning

Base backbone models often lack the ability to invoke tools reliably, even when explicit tool descriptions are provided. In preliminary experiments, models frequently rely on pure natural-language reasoning and fail to issue tool calls when file access or computation is required, making direct reinforcement learning ineffective.

To address this, we perform a cold-start supervised fine-tuning stage to bootstrap basic tool-use competence. The model is trained on the verified task-solution pairs generated by our pipeline (Section 3). These trajectories interleave reasoning, tool calls, tool responses, and final answers, providing high-quality demonstrations of correct file operations. This stage is not intended to improve task accuracy, but to ensure that the model can operate within the sandbox environment, allowing subsequent reinforcement learning to focus on improving reasoning quality and strategy selection.

### 4.3. Reinforcement Learning in a Persistent File Environment

While supervised fine-tuning teaches the model the basic syntax of tool invocation, it falls short of enabling robust planning in dynamic environments. To bridge this gap, we formalize file reasoning as a sequential decision-making process grounded in a persistent sandbox. More details regarding the implementation can be found in Appendix A.

**Problem Formulation.**  We define a reasoning trajectory as a sequence $\tau = \{s_0, a_0, o_0, \ldots, s_T, a_T, o_T\}$, where at each step $t$:

- $s_t$ denotes the **agent state**, comprising the dialogue history and the current visual/textual observations.

- $a_t$ represents the **action**, specifically a generated thought trace followed by an executable code block or shell command.

- $o_t$ is the **execution observation** returned by the environment (e.g., standard output, error trace, or file artifacts).

Crucially, distinct from stateless tool-use benchmarks, the execution of $a_t$ modifies a latent environment context $\mathcal{E}$. For

instance, an action might involve unzipping a compressed archive to reveal a directory structure, or extracting specific data from a PDF and saving it as an intermediate CSV for further analysis. These operations do not merely return a value but permanently alter the disk state and file availability. Since the file state inherits all prior modifications, the agent must continuously update its understanding of the environment to plan effective future moves.

**Environment Setup and Feedback.** We implement this formulation using a containerized sandbox loop, as detailed in Algorithm 1. This setup ensures strict persistence within each rollout. For a given task, the raw source files are pre-loaded into the environment context $\mathcal{E}_0$. When the policy $\pi_\theta(a_t|s_t)$ generates a tool call, it is executed synchronously against $\mathcal{E}_t$, producing real-time feedback $o_t$. This exposes the model to genuine execution dynamics, such as `FileNotFoundError` or shape mismatches in dataframes, providing a dense signal for learning error recovery and adaptive planning.

**Optimization Objective.** We optimize the policy $\pi_\theta$ to maximize the expected return over trajectories. Given the diversity of valid solution paths (e.g., using "pandas" vs. "csv" libraries), we employ a sparse, outcome-oriented reward function $R(\tau)$ rather than step-level supervision:

$$R(\tau) = \mathbb{I}(y_{\text{pred}} = y_{\text{gt}}) + \lambda \cdot \mathbb{I}(\text{invalid\_format}) \quad (1)$$

where $y_{\text{pred}}$ is the final answer extracted from $\tau$, and $y_{\text{gt}}$ is the ground truth. We set $\lambda = -0.2$ to strictly penalize structural errors (e.g., XML tag mismatches) without explicitly rewarding basic compliance. We use Group Relative Policy Optimization (GRPO) to update the model (Shao et al., 2024), encouraging it to discover robust strategies that navigate the latent environment state $\mathcal{E}$ to reach the correct solution.

## 5. Evaluating Model Performance

In this section, we evaluate diverse LLMs on the benchmark constructed in Section 3.5 and compare its difficulty against existing datasets. This comparison demonstrates that our real-world, unprocessed file tasks pose significant challenges, remaining a rigorous testbed even for state-of-the-art proprietary models.

### 5.1. Evaluation Setup

Our evaluation follows exactly the same sandbox-based rollout logic used during training (Section 4.3). Each model is executed inside the unified, stateful sandbox with real code and shell interactions, ensuring strict comparability and full reproducibility across model families. We conduct large-scale benchmarking across a diverse set of

---

**Algorithm 1** Stateful Reinforcement Learning in Persistent Sandbox

1: **Input:** dataset $\mathcal{D}$, policy $\pi_\theta$, tools $\mathcal{T}$, group size $G$
2: **Output:** fine-tuned policy $\pi_\theta^\star$
3: **for** each epoch **do**
4: $\quad B \leftarrow \mathcal{D}.\text{SAMPLEBATCH}()$
5: $\quad$ **for** $x \in B$ **in parallel do**
6: $\quad\quad p_x \leftarrow x.\text{prompt}, F_x \leftarrow x.\text{prewrites}$
7: $\quad\quad$ **for** $k = 1, \ldots, G$ **do**
8: $\quad\quad\quad \tau_{x,k} \leftarrow \text{ROLLOUT}(p_x, F_x, \pi_\theta)$
9: $\quad\quad\quad r_{x,k} \leftarrow \text{COMPUTEREWARD}(\tau_{x,k}, x.\text{answer})$
10: $\quad\quad$ **end for**
11: $\quad\quad \hat{A}_{x,1:G} \leftarrow \text{GROUPNORMALIZE}(r_{x,1:G})$
12: $\quad$ **end for**
13: $\quad \theta \leftarrow \text{GRPOUPDATE}(\theta, \{\tau_{x,k}\}, \{\hat{A}_{x,k}\})$
14: **end for**
15: **return** $\pi_\theta^\star$
16: **Function** ROLLOUT$(p, F, \pi)$:
17: Initialize chat with $p$; $\tau \leftarrow \emptyset$; sid $\leftarrow$ None
18: **while** not terminated **do**
19: $\quad s_t \leftarrow$ chat; $a_t \leftarrow \pi(s_t)$; parse calls $C_t$ from $a_t$
20: $\quad$ **if** $C_t = \emptyset$ **then**
21: $\quad\quad \tau \leftarrow \tau \cup \{(s_t, a_t, \emptyset)\}$; **break**
22: $\quad$ **end if**
23: $\quad$ **if** sid $=$ None **then**
24: $\quad\quad$ sid $\leftarrow$ CREATESANDBOX(); PREWRITE-FILES(sid, $F$)
25: $\quad$ **end if**
26: $\quad o_t \leftarrow \text{EXECUTETOOLS}(\text{sid}, C_t, \mathcal{T})$
27: $\quad$ Append $o_t$ to chat; $\tau \leftarrow \tau \cup \{(s_t, a_t, o_t)\}$
28: **end while**
29: **if** sid $\neq$ None **then**
30: $\quad$ DESTROYSANDBOX(sid)
31: **end if**
32: **return** $\tau$

---

models, including open-source models such as Llama-3.1-8B (AI@Meta, 2024), Qwen2.5-32B, Qwen2.5-72B (Qwen Team, 2024), Qwen3 (Yang et al., 2025), GLM-5 (GLM-5-Team et al., 2026), Kimi-K2-Thinking (Team et al., 2025), and DeepSeek-V3.1 (DeepSeek-AI et al., 2025), as well as proprietary models such as GPT-4.1, Gemini-2.5-Pro (Comanici et al., 2025), o3 (OpenAI, 2025b), Grok-4 (xAI, 2025) and GPT-5 (OpenAI, 2025a). All models are evaluated under identical tool-augmented execution constraints, including the persistent sandbox, real file operations, and the fixed 15-round interaction budget.

### 5.2. Main Results

Table 2 presents the performance of different model families across four file types, including XLSX, PDF, PPTX, and DOCX. In addition to accuracy on each file type, we also

*Table 2.* Comparison of model performance across four file types (XLSX, PDF, PPTX, DOCX) under the same evaluation setting. Models are organized into Closed-source, Open-source, and our **FIRE** models. We highlight the best and second-best performance within the **Closed-source** group and the combined **Open-source + FIRE** group in **bold** and underlined, respectively.

| Model | XLSX Acc | PDF Acc | PPTX Acc | DOCX Acc | Overall Acc | Overall TC |
|---|---|---|---|---|---|---|
| **Closed-source Models** | | | | | | |
| GPT-4.1 | 18.03 | 31.03 | 22.22 | 34.83 | 24.15 | 6.26 |
| Gemini-2.5-Pro | 23.47 | 34.10 | 42.78 | 52.24 | 33.25 | 5.03 |
| Grok-4 | **53.91** | 66.28 | **67.22** | 68.66 | **60.89** | 5.92 |
| GPT-5 | 47.28 | **72.41** | 66.67 | **76.12** | 60.16 | 5.00 |
| **Open-source Models** | | | | | | |
| Llama-3.1-8B-Instruct | 5.61 | 1.15 | 7.22 | 0.00 | 3.98 | 6.08 |
| Qwen2.5-7B-Instruct | 5.61 | 1.53 | 9.44 | 2.49 | 4.80 | 4.56 |
| Qwen3-8B | 8.16 | 7.28 | 6.11 | 1.99 | 6.67 | 4.36 |
| Qwen2.5-32B-Instruct | 7.14 | 5.75 | 16.11 | 6.47 | 8.05 | 4.73 |
| Qwen3-32B | 10.03 | 8.81 | 17.78 | 11.44 | 11.14 | 4.63 |
| Qwen2.5-72B-Instruct | 8.50 | 12.64 | 24.44 | 14.93 | 12.76 | 6.62 |
| Kimi-K2-Thinking | 35.03 | 60.92 | 58.89 | 59.20 | 47.97 | 8.92 |
| DeepSeek-V3.1 | 32.82 | 59.00 | 50.56 | 54.73 | 44.55 | 6.79 |
| GLM-5 | **50.51** | **76.63** | **71.11** | **76.62** | **63.33** | 6.34 |
| **Our FIRE Models** | | | | | | |
| Qwen2.5-7B-FIRE (SFT) | 21.60 | 25.29 | 26.67 | 18.91 | 22.68 | 5.13 |
| Qwen2.5-7B-FIRE (RL) | 26.70 | 36.78 | 31.11 | 34.83 | 30.81 | 5.55 |
| Qwen2.5-32B-FIRE (SFT) | 34.69 | 50.19 | 42.22 | 43.78 | 40.57 | 6.81 |
| Qwen2.5-32B-FIRE (RL) | 39.63 | 65.13 | 50.00 | 51.74 | 48.54 | 6.12 |
| Qwen3-32B-FIRE (SFT) | 32.99 | 54.79 | 46.67 | 52.24 | 42.76 | 7.43 |
| Qwen3-32B-FIRE (RL) | 40.14 | 66.28 | 53.33 | 56.72 | 50.33 | 6.49 |

report the overall accuracy and the average number of tool calls, which reflect both task-solving effectiveness and interaction efficiency. For each model, we report the average accuracy over three independent runs. Beyond these raw results, we highlight three critical observations regarding the impact of our framework:

**RL Improves Reasoning Efficiency via Strategic Planning.** A naive expectation might be that reinforcement learning improves accuracy by encouraging the model to attempt more steps or retries. However, our results indicate the opposite: RL consistently reduces the average number of tool calls while increasing accuracy. For instance, **Qwen3-32B-FIRE** improves overall accuracy from 42.76% (SFT) to 50.33% (RL) while reducing the average number of tool calls from 7.43 to 6.49. This suggests that the model learns to use interaction rounds more selectively, condensing file inspection, evidence extraction, and computation into shorter trajectories. To validate this, we conduct a qualitative analysis of reasoning trajectories in Appendix I. As detailed in the case studies, RL induces two key behavioral shifts: (1) more efficient recovery from structural or execution-level obstacles, such as merged cells in spreadsheets, without repeatedly reinspecting the same evidence; and (2) more

targeted tool use, where the model applies programmatic parsing to extract and filter task-relevant evidence rather than relying on brittle, broad shell-style searches.

**Performance Variance Across File Structures.** Our benchmark reveals a distinct hierarchy in difficulty across file types, reflected in the performance of top-tier models. XLSX and PPTX tasks remain more challenging overall than text-dominant formats such as PDF and DOCX.

This disparity highlights the varying nature of reasoning required: XLSX tasks demand rigorous quantitative reasoning and complex code logic (e.g., multi-step pandas operations), while PPTX tasks require navigating hierarchical object structures such as slides, shapes, and embedded tables. In contrast, PDF and DOCX tasks, which more often rely on linear information extraction, generally lead to higher accuracy for stronger models. This confirms that our benchmark captures the structural difficulties of spreadsheet and presentation workflows, rather than merely testing reading comprehension.

**Narrowing the Gap with Proprietary Models.** Under identical sandbox constraints, our Qwen3-32B-FIRE

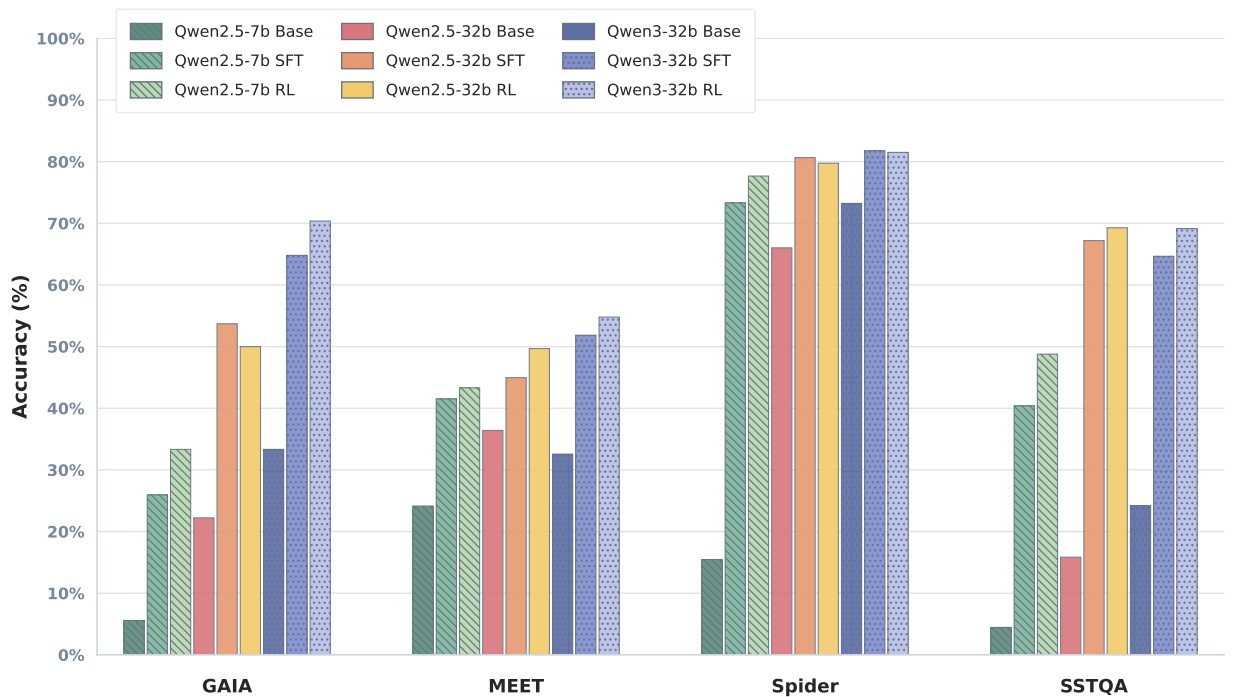

*Figure 2.* Performance of Base, SFT, and RL models on out-of-distribution benchmarks. Overall, SFT provides substantial improvements over the base models, and RL often brings further gains, especially for the Qwen3-32B backbone.

(RL) model reaches 50.33% overall accuracy, substantially outperforming its base counterpart Qwen3-32B (11.14%) and approaching strong proprietary models such as GPT-5 (60.16%) and Grok-4 (60.89%). It also outperforms DeepSeek-V3.1 (44.55%) and Kimi-K2-Thinking (47.97%), despite using a substantially smaller 32B backbone than many frontier-scale systems. This is notable because many general-purpose open-source models, although strong in language understanding and reasoning, are not explicitly trained for long-horizon agentic interaction with real files. As a result, they often struggle to adapt to sandbox-based workflows that require repeated tool use, execution feedback, error recovery, and persistent file-state management. By contrast, our stateful reinforcement learning framework directly exposes the model to these interaction dynamics during training, enabling it to learn when to inspect files, how to choose appropriate tools, and how to revise its strategy after failed executions. These findings show that targeted stateful reinforcement learning can make moderate-size open-source models substantially more competitive on complex agentic workflows, even though a gap remains to the strongest frontier models such as GLM-5, Grok-4, and GPT-5.

### 5.3. Comparison with Existing Benchmarks

To contextualize the observed results, we evaluate a representative subset of models on several existing file-based or table-oriented benchmarks, including Spider, SSTQA,

MEET, and GAIA. All evaluations are performed under the same unified sandbox environment.[1] As shown in Table 3, model accuracy on our benchmark is consistently lower than on prior datasets. Even GPT-5 performs substantially better on SSTQA and GAIA than on ours. This gap suggests that, under the same evaluation setting and tool-based interface, our benchmark poses a more challenging test of file-based agent capabilities than existing datasets.

## 6. Analysis

### 6.1. Generalization to Out-of-Distribution Data

We further evaluate our trained models on out-of-distribution benchmarks to assess their generalization ability. As shown in Figure 2, the performance follows the same upward trend observed in the in-distribution setting. SFT yields a substantial initial improvement, while RL often provides additional gains across the evaluated OOD benchmarks. These improvements on OOD benchmarks suggest that our training corpus effectively captures diverse real-world file structures. Built from CommonCrawl-sourced documents and processed through a unified pipeline, the corpus exposes the model to a wide range of file formats and organizational patterns. This diversity enables the model to

---

[1]For GAIA, we evaluate only the validation subset that contains file-related tasks to ensure a fair comparison. For MEET, we evaluate using the publicly accessible released files.

*Table 3.* Performance of representative models across benchmarks (accuracy, %).

| Model | Spider | SSTQA | MEET | GAIA | Ours |
|---|---|---|---|---|---|
| GPT-5 | 85.2 | 76.7 | 72.8 | 77.8 | 60.2 |
| Gemini-2.5 | 82.4 | 56.2 | 48.3 | 66.7 | 33.3 |
| DS-V3.1 | 84.4 | 74.1 | 56.7 | 72.2 | 44.6 |
| GLM-5 | 85.6 | 79.2 | 70.9 | 87.5 | 63.3 |
| o3 | 71.3 | 51.7 | 49.4 | 61.1 | 26.3 |
| **Avg.** | 81.8 | 67.6 | 59.6 | 73.1 | 45.5 |

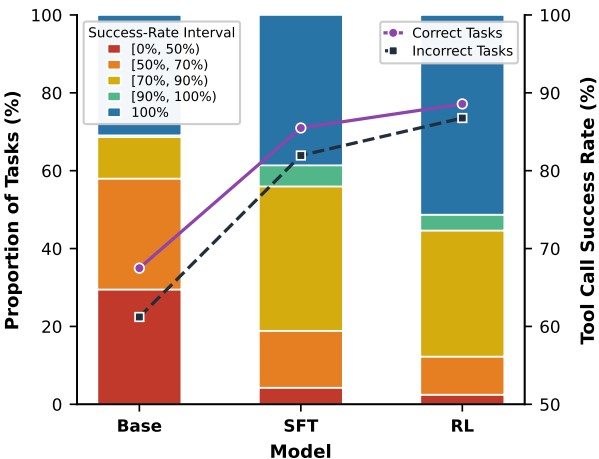

*Figure 3.* Distribution of tasks across tool call success rate intervals (stacked bars) and average tool call success rates for correct vs. incorrect tasks (lines) across Base, SFT, and RL models. Model performance improves progressively through the three training stages: from Base to SFT to RL.

learn transferable and robust tool-use strategies, rather than overfitting to dataset-specific artifacts. Consequently, the combined SFT and RL training significantly enhances both in-domain performance and cross-domain generalization.

### 6.2. Analysis of Tool-Calling Behaviors

To understand the model's tool-use patterns, we compare the Base, SFT, and RL models in Figure 3. First, we observe a consistent improvement in tool-call success rates from Base to SFT, and further to RL, confirming that each training stage improves execution reliability. Second, while correctly answered tasks still exhibit somewhat higher tool success rates in the Base and SFT models, this gap becomes negligible in the RL model. This suggests that intermediate tool failures are not the sole determinant of final task correctness: the model can often recover from execution errors by using the returned feedback to revise subsequent actions. Based on this observation, we design the reward function to avoid directly penalizing intermediate tool failures, encouraging exploration and recovery rather than overly conservative behavior.

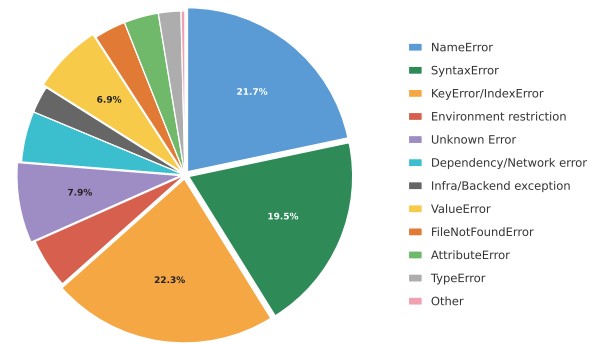

*Figure 4.* Distribution of file-related error types encountered in file reasoning tasks.

### 6.3. File Error Analysis

Figure 4 shows the distribution of tool-execution error types in our tasks. Among the observed errors, schema-related failures such as `KeyError` and `IndexError` form the largest category, suggesting that models often struggle to ground their operations in the actual structure of parsed files. These errors commonly arise when the agent assumes clean headers, stable sheet names, regular table layouts, or consistently named fields in noisy spreadsheets and document-derived tables. `NameError` is also a major category, indicating that models still struggle to maintain consistent execution state across multiple rounds of tool use, for example by reusing variables, imports, or intermediate results that were defined in earlier attempts but are missing in the current code block. `SyntaxError` appears at a comparable frequency, usually caused by invalid generated code, such as mixing shell-style commands into Python execution blocks or emitting malformed command strings. In addition, value-related, dependency, environment, and backend failures remain non-negligible, often triggered by malformed data assumptions, package installation attempts, external downloads, unavailable system commands, or infrastructure-level interruptions. We provide a model-wise breakdown of these error profiles in Appendix E.

### 6.4. Analysis of Turn Budget and Tool Call Frequency

We further analyze the trade-off between reasoning depth and inference efficiency. Figure 5 shows the cumulative accuracy as a function of the maximum interaction round budget for four representative models: GPT-5, GLM-5, Gemini-2.5-Pro, and Qwen3-RL. Across all four models, the majority of the accuracy gain is obtained within the first 15 rounds, while the marginal improvement beyond 15 rounds is limited. These results suggest that increasing the turn budget beyond 15 substantially increases inference cost and

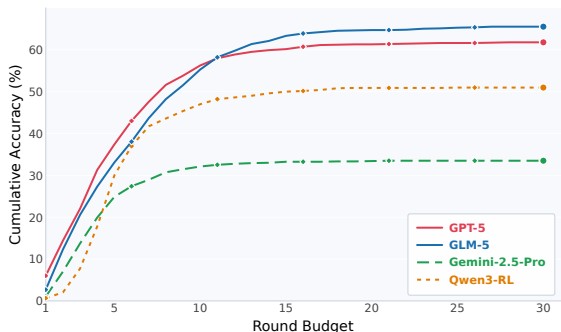

*Figure 5.* Cumulative accuracy as a function of the maximum interaction round budget for GPT-5, GLM-5, Gemini-2.5-Pro, and Qwen3-RL. Most performance gains are achieved within the first 15 rounds, while extending the budget to 30 rounds yields only limited additional improvement.

latency, but yields only minor accuracy gains. Therefore, considering both efficiency and monetary cost, we choose a maximum turn budget of 15 in our main experiments.

We also examine how performance varies with the number of tool calls for different file types, as shown in Figure 12. Overall, more tool calls do not necessarily translate into better performance. Across Base, SFT, and RL models, the highest correct counts are generally concentrated within a relatively small number of tool calls, although the exact peak varies across file types and training stages. This pattern suggests that successful trajectories often rely on concise and targeted file inspection rather than prolonged interaction. Beyond this moderate range, additional tool interactions tend to provide limited gains and may instead reflect redundant exploration, repeated recovery from earlier execution errors, or inefficient search. Together, these observations support using a moderate interaction budget and avoiding unnecessarily long tool-use trajectories.

## 7. Conclusion

In this work, we formalized the File Reasoning setting, shifting the paradigm from processing flattened text to interacting with raw, stateful file environments. Our high-difficulty benchmark reveals that even state-of-the-art models struggle with the structural ambiguity of real-world documents, a capability gap often masked by traditional pre-processing. To bridge this gap, we developed a reinforcement learning framework grounded in a persistent sandbox, enabling agents to learn robust planning strategies directly from genuine execution feedback. We trained **FIRE**, a family of models that demonstrate consistent gains in both reasoning accuracy and tool-use efficiency. Notably, **Qwen3-32B-FIRE** achieves the strongest performance among open-source models of comparable scale, demonstrating that environment-grounded training is a more effective path to

robust agency than scale alone. We hope this work paves the way for future agents capable of autonomously navigating the complex information landscape of the real world.

## Impact Statement

This paper presents work whose goal is to advance the field of machine learning by improving language model agents' ability to reason over and interact with real-world files. We expect this research to enable more reliable and practical AI systems for information processing tasks, including document analysis, spreadsheet reasoning, and automated report understanding. Since file-based agents may be applied to sensitive or proprietary documents, practical deployment should include appropriate access control, sandbox isolation, privacy-preserving data handling, and human oversight in high-stakes settings. With these safeguards in place, we do not expect our work to introduce significant additional negative societal impacts.

## Acknowledgments

This paper is supported by NSFC project 62476009. We thank the reviewers for the valuable comments and suggestions, which help us clarify the motivation and improve the presentation of our work.

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

# A. Implementation Details for RL Framework

## A.1. Infrastructure and Agent Loop

Our training infrastructure is built upon the `verl` framework, a distributed reinforcement learning library optimized for LLMs. The agent interaction loop is designed as a finite state machine that interleaves autoregressive token generation with synchronous tool execution. Specifically, the model generates tokens until a tool-call sequence (formatted in the Hermes schema) is detected. The system then pauses generation, parses the structured arguments, executes the tool in the sandbox, and appends the output to the context window. This cycle continues until the model produces a final answer or reaches the maximum turn limit ($T = 15$). To ensure correct credit assignment, we apply a binary mask to the training sequences: tokens generated by the policy model are marked with 1, while tokens injected by the environment (i.e., tool outputs and system prompts) are marked with 0. This ensures that the policy gradient is computed strictly over the agent's decisions.

## A.2. Persistent Sandbox Environment

A critical component of our framework is the management of persistent file states. Unlike stateless function calls, our environment maintains a consistent container session throughout a full trajectory.

- **Lifecycle Management:** A unique sandbox session is initialized at the first tool call of a trajectory. All subsequent Python code execution and Shell commands within the same rollout are routed to this session, ensuring that file system modifications (e.g., creating intermediate data files) are preserved across turns. The session is strictly isolated and destroyed only after the trajectory terminates.

- **File Pre-loading:** For tasks involving existing files (e.g., analyzing an Excel sheet), we implement a pre-write mechanism that injects the required files into the sandbox working directory via base64 encoding prior to the first execution step.

- **Tool Specifications:** We expose two primary tools:
  - `ExecuteCode`: Runs Python scripts with a 60-second timeout. The environment comes pre-installed with standard data science libraries (pandas, numpy, matplotlib, etc.).
  - `ExecuteShell`: Executes Bash commands to support file navigation and text processing (e.g., `grep`, `ls`, `mv`).

## A.3. Reward Computation

As formalized in Equation (1) of the main text, our reward function consists of a correctness verification term and a structural penalty:

$$R(\tau) = \underbrace{\mathbb{I}(y_{\text{pred}} = y_{\text{gt}})}_{\text{Correctness Reward}} + \lambda \cdot \underbrace{\mathbb{I}(\text{invalid\_format})}_{\text{Format Penalty}}$$

- **Correctness Verification** ($\mathbb{I}(y_{\textbf{pred}} = y_{\textbf{gt}})$)**:** We employ a pure **LLM-as-a-Judge** approach to verify the final answer. Specifically, we use **GPT-5-mini** as the evaluator. For each trajectory, the judge compares the model's prediction $y_{\text{pred}}$ against the ground truth $y_{\text{gt}}$ based on semantic equivalence and returns a binary decision (1 for correct, 0 for incorrect). This allows for robust verification across diverse output formats (e.g., handling floating-point precision differences or varying text phrasing) without relying on rigid string matching rules.

- **Format Penalty** ($\lambda$)**:** We set the penalty coefficient $\lambda = -0.2$. The term $\mathbb{I}(\text{invalid\_format})$ is set to 1 if the trajectory fails to follow the required XML schema (e.g., missing `<think>` tags or unclosed `<tool_call>` blocks), and 0 otherwise.

## A.4. Training Configuration

We initialize training from supervised fine-tuned checkpoints of Qwen2.5-7B, Qwen2.5-32B, and Qwen3-32B. Training is conducted using Group Relative Policy Optimization (GRPO) on 8 nodes with 8 GPUs each (64 GPUs total). We adopt a clipped objective with clip_ratio_low = 0.20, clip_ratio_high = 0.28, and clip_ratio_c = 10.0. We use vLLM for high-throughput inference during rollout generation.

# B. Examples in Benchmark

We illustrate representative examples from the four file types in our benchmark. For brevity, only the portions directly related to the question are displayed.

---

### Example: Eating & Health Time Use (EXCEL)

**Benchmark Question.** *Define $S = $ (Food preparation and cleanup, mean $+$ Primary eating and drinking, mean) $-$ (Paid work, mean $+$ Travel, mean). Considering only Gender 'Men' and 'Women', for each gender find the **Amount** with the maximum $S$; if tied on $S$, choose the tied **Amount** with the smallest (Food preparation and cleanup, SE $+$ Primary eating and drinking, SE), and if still tied, pick the lexicographically smallest **Amount**. Finally, between the two gender-specific selections, choose the one with the larger $S$; if equal, break ties by lexicographic order of **Gender**.*

**Challenge Analysis.** The sheet is hierarchical: under *By gender* there are **Men/Women**, each with multiple **Amount** levels. For each activity (Food preparation and cleanup / Primary eating and drinking / Paid work / Travel), columns split into *Mean* and *Standard error (SE)*. Solving requires correctly mapping the multi-row header to tidy fields {Gender, Amount, Activity, Metric}, selecting only Gender $\in$ {Men, Women}, computing $S$ per (Gender, Amount) from *Mean* columns, and applying tie-breaking by (Food prep SE $+$ Primary eating SE) and then lexicographic Amount. Finally compare the two gender winners by $S$ (then lexicographic Gender if equal). The core difficulty is abstracting the logical grouping from a visually merged, multi-level header and aligning the correct Mean/SE pairs.

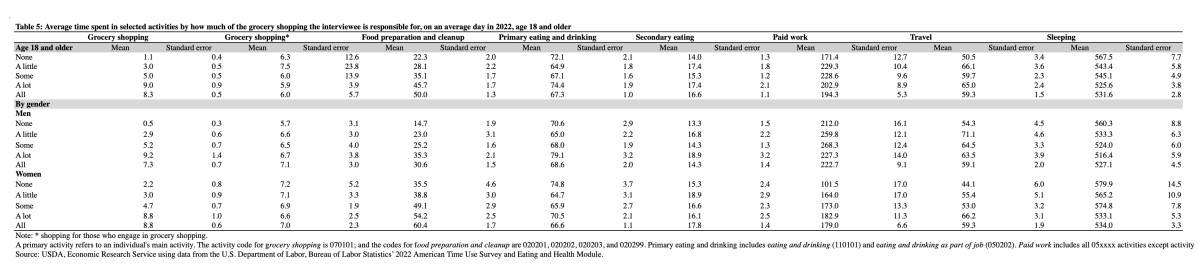

**Table 5: Average time spent in selected activities by how much of the grocery shopping the interviewee is responsible, on an average day in 2022, age 18 and older**

| Age 18 and older | Grocery shopping | | Grocery shopping* | | Food preparation and cleanup | | Primary eating and drinking | | Secondary eating | | Paid work | | Travel | | Sleeping | |
| --- | --- | --- | --- | --- | --- | --- | --- | --- | --- | --- | --- | --- | --- | --- | --- | --- |
| | Mean | Standard error | Mean | Standard error | Mean | Standard error | Mean | Standard error | Mean | Standard error | Mean | Standard error | Mean | Standard error | Mean | Standard error |
| None | 1.1 | 0.4 | 6.3 | 12.6 | 22.3 | 2.0 | 72.1 | 2.1 | 14.0 | 1.3 | 171.4 | 12.7 | 50.5 | 3.4 | 567.5 | 7.7 |
| A little | 3.0 | 0.5 | 7.5 | 23.8 | 28.1 | 2.2 | 64.9 | 1.8 | 17.4 | 1.8 | 229.3 | 10.4 | 66.1 | 3.6 | 543.4 | 5.8 |
| Some | 5.0 | 0.5 | 6.0 | 13.9 | 35.1 | 1.7 | 67.1 | 1.6 | 15.3 | 1.2 | 228.6 | 9.6 | 59.7 | 2.3 | 545.1 | 4.9 |
| A lot | 9.0 | 0.9 | 5.9 | 3.9 | 45.7 | 1.7 | 74.4 | 1.9 | 17.4 | 2.1 | 202.9 | 8.9 | 65.0 | 2.4 | 525.6 | 3.8 |
| All | 8.3 | 0.5 | 6.0 | 5.7 | 50.0 | 1.3 | 67.3 | 1.0 | 16.6 | 1.1 | 194.3 | 5.3 | 59.3 | 1.5 | 531.6 | 2.8 |
| **By gender** | | | | | | | | | | | | | | | | |
| **Men** | | | | | | | | | | | | | | | | |
| None | 0.5 | 0.3 | 5.7 | 3.1 | 14.7 | 1.9 | 70.6 | 2.9 | 13.3 | 1.5 | 212.0 | 16.1 | 54.3 | 4.5 | 560.3 | 8.8 |
| A little | 2.9 | 0.6 | 6.6 | 3.0 | 23.0 | 3.1 | 65.0 | 2.2 | 16.8 | 2.2 | 259.8 | 12.1 | 71.1 | 4.6 | 533.3 | 6.3 |
| Some | 5.2 | 0.7 | 6.5 | 4.0 | 25.2 | 1.6 | 68.0 | 1.9 | 14.3 | 1.3 | 268.3 | 12.4 | 64.5 | 3.3 | 524.0 | 6.0 |
| A lot | 9.2 | 1.4 | 6.7 | 3.8 | 35.3 | 2.1 | 79.1 | 3.2 | 18.9 | 3.2 | 227.3 | 14.0 | 63.5 | 3.9 | 516.4 | 5.9 |
| All | 7.3 | 0.7 | 7.1 | 3.0 | 30.6 | 1.5 | 68.6 | 2.0 | 14.3 | 1.4 | 222.7 | 9.1 | 59.1 | 2.0 | 527.1 | 4.5 |
| **Women** | | | | | | | | | | | | | | | | |
| None | 2.2 | 0.8 | 7.2 | 5.2 | 35.5 | 4.6 | 74.8 | 3.7 | 15.3 | 2.4 | 101.5 | 17.0 | 44.1 | 6.0 | 579.9 | 14.5 |
| A little | 3.0 | 0.9 | 7.1 | 3.3 | 38.8 | 3.0 | 64.7 | 3.1 | 18.9 | 2.9 | 164.0 | 17.0 | 55.4 | 5.1 | 565.2 | 10.9 |
| Some | 4.7 | 0.7 | 6.9 | 1.9 | 49.1 | 2.9 | 65.9 | 2.7 | 16.6 | 2.3 | 173.0 | 13.3 | 53.0 | 3.2 | 574.8 | 7.8 |
| A lot | 8.8 | 1.0 | 6.6 | 2.5 | 54.2 | 2.5 | 70.5 | 2.1 | 16.1 | 2.5 | 182.9 | 11.3 | 66.2 | 3.1 | 533.1 | 5.3 |
| All | 8.8 | 0.6 | 7.0 | 2.3 | 60.4 | 1.7 | 66.6 | 1.1 | 17.8 | 1.4 | 179.0 | 6.6 | 59.3 | 1.9 | 534.0 | 3.3 |

Note: * shopping for those who engage in grocery shopping.
A primary activity refers to an individual's main activity. The activity code for *grocery shopping* is 070101; and the codes for *Food preparation and cleanup* are 020201, 020202, 020203, and 020299. Primary eating and drinking includes *eating and drinking* (110101) and *eating and drinking as part of job* (050202). *Paid work* includes all 05xxxx activities except activity
Source: USDA, Economic Research Service using data from the U.S. Department of Labor, Bureau of Labor Statistics' 2022 American Time Use Survey and Eating and Health Module.

*Figure 6.* Example real-world EXCEL file used in our benchmark.

**FINAL ANSWER:** <Women, None> ✓

## Example: Medical Meeting Report

**Benchmark Question.** *Calculate the sum of two values: (1) the patient list size at the time of the last meeting, and (2) the average wasted time in minutes per individual GP 'Did Not Attend' (DNA) appointment across the period January to March 2025. Provide the final result as a whole number.*

**Challenge Analysis.** This task requires (i) locating the current List Size value 7917, identifying the reported decrease of 32 since the last meeting, and reconstructing the patient list size at the time of the last meeting as $7917 + 32 = 7949$; and (ii) navigating the *Appointments/DNA Rates ("Did Not Attend") and telephone appointments generally* section, understanding the month-wise table layout (January, February, March 2025), computing the *average wasted time in minutes per individual GP DNA* across the three months, and finally summing that average with the reconstructed historical list size. The model must correctly interpret mixed text–table structure, annotated cells, and avoid double-counting.

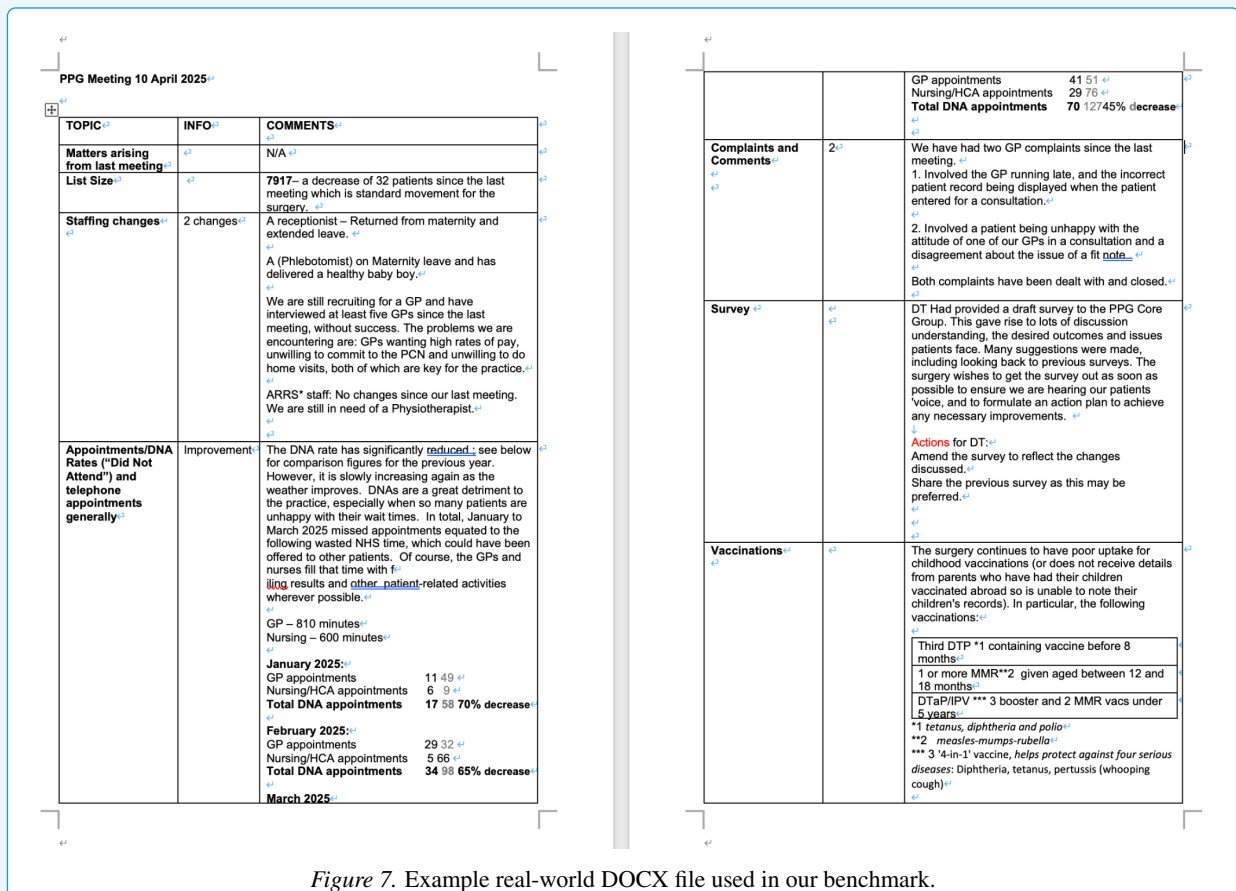

*Figure 7.* Example real-world DOCX file used in our benchmark.

**FINAL ANSWER:** 7959 ✓

**Example: Workforce Equality Report (PDF)**

**Benchmark Question.** *What is the result of multiplying the total number of employees on reduced pay (and thus excluded from the pay quartile data) by the number of distinct action themes that have at least one action with a review date in April 2021?*

**Challenge Analysis.** The main difficulty of this question lies in integrating information across multiple sections and avoiding confusion. You first need to extract the total workforce and quartile counts from different parts of the report to correctly compute the number of employees on reduced pay. Then, in the "Action and Review Date" table, you must identify the four distinct themes and carefully determine which have at least one action reviewed in April 2021—without mistakenly counting individual actions as separate themes. The challenge is not the arithmetic itself but accurately understanding the document's structure and logical hierarchy.

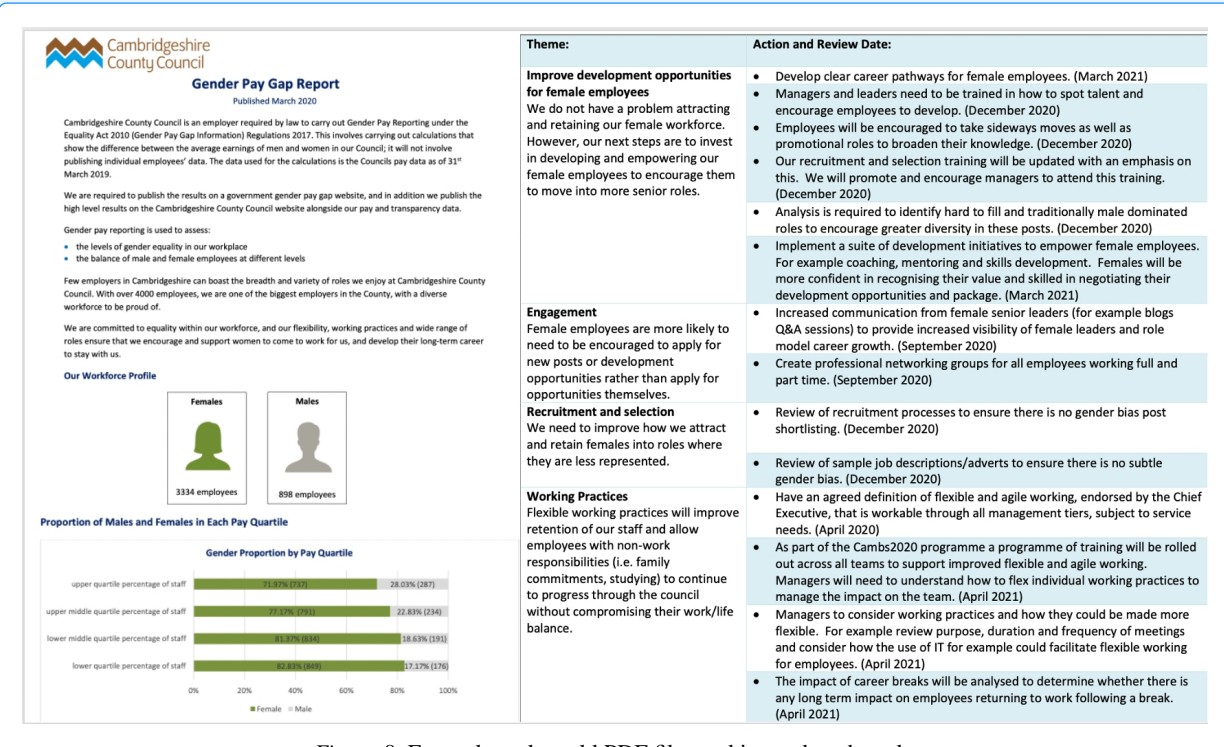

*Figure 8.* Example real-world PDF file used in our benchmark.

**FINAL ANSWER:** <133> ✓

**Example: National Energy Potential Report (PPTX)**

**Benchmark Question.** *Which Brazilian state has the highest technical energy potential (*Potencial Energético por UF*) as reported in the table titled "Brasil: Potencial Energético por UF (em GW)"? What is the corresponding potential value (in gigawatts) for that state?*

**Challenge Analysis.**
The main difficulty lies in correctly locating and interpreting the chart titled "Brasil: Potencial Energético por UF (em GW)" inside the native presentation file. Although the values are visually presented as a chart, the relevant state labels and numeric potentials are stored in the PPTX chart structure rather than as a flat table. The model must recover the association between each UF label and its corresponding *Potencial Técnico*, understand that "UF" refers to Brazilian states, and then identify the state with the maximum value rather than relying on approximate visual reading.

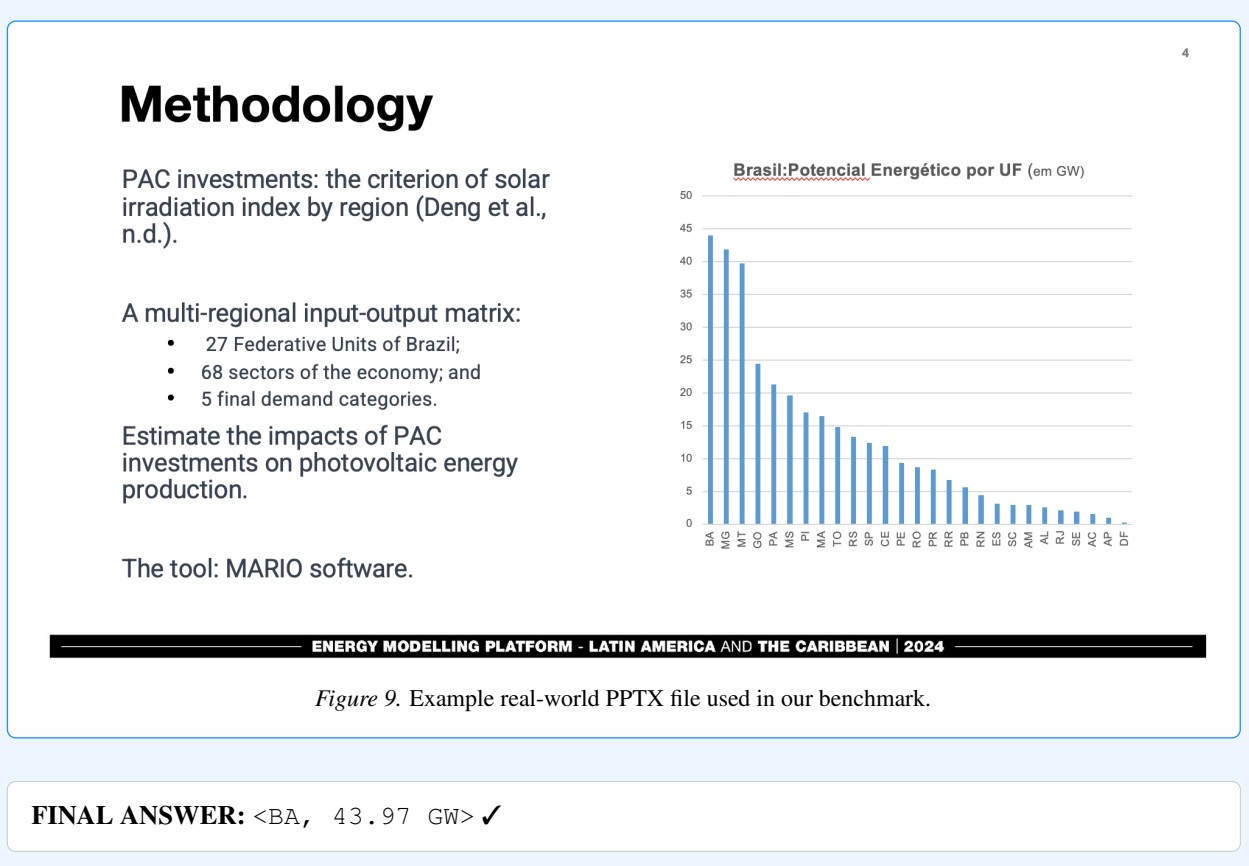

*Figure 9.* Example real-world PPTX file used in our benchmark.

**FINAL ANSWER:** `<BA, 43.97 GW>` ✓

## C. Data Contamination Check

Since most files in our benchmark are collected from public web sources, there is a potential risk that some models may have seen related content during pre-training or post-training. To examine this risk, we conduct a no-file setting, where the input files are removed and models are asked to answer the questions based only on their parametric knowledge. Under this setting, model performance drops substantially. Specifically, GLM-5 achieves an accuracy of 6.34% and Kimi-K2-Thinking achieves 4.88%. These results show that, without access to the files, models can answer only a very small fraction of the benchmark questions. Therefore, the main evaluation signal of our benchmark comes from models' actual understanding and use of file contents, rather than from memorized answers.

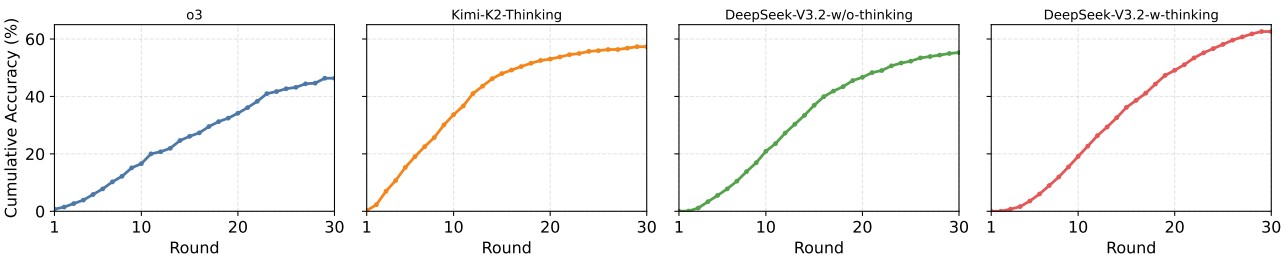

*Figure 10.* Cumulative accuracy as a function of the maximum interaction round budget for four additional models: o3, Kimi-K2-Thinking, DeepSeek-V3.2-w/o-thinking, and DeepSeek-V3.2-w-thinking.

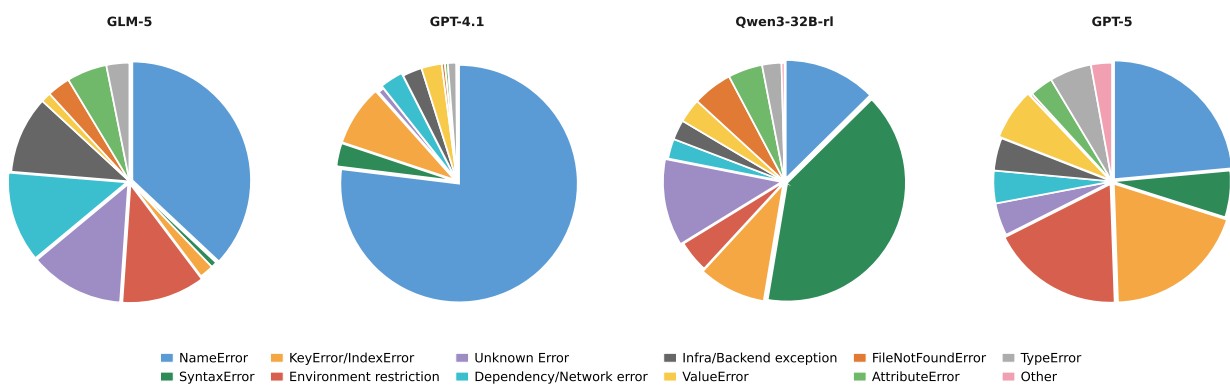

*Figure 11.* Distribution of tool-execution error types for four representative models in file reasoning tasks.

## D. Extended Analysis of Interaction Budget

Figure 10 shows the cumulative accuracy curves for four additional models with a maximum budget of 30 interaction rounds: o3, Kimi-K2-Thinking, DeepSeek-V3.2-w/o-thinking, and DeepSeek-V3.2-w-thinking. Compared with the representative models discussed in the main text, these models show more pronounced late-round improvements beyond round 15. A closer inspection of the late-round successful cases suggests that the extra gains often come from recovering earlier execution failures rather than from more efficient task completion. In many such cases, the additional rounds are spent on repeated file inspection, re-importing libraries, redefining variables or file paths, rerunning nearly identical extraction code, and repairing unsuccessful intermediate tool interactions before the model finally converges to the correct answer. This pattern indicates that a larger round budget can improve final accuracy for some models, but the improvement is typically achieved through longer and less efficient interaction trajectories rather than through faster or more stable tool use.

## E. Tool-Execution Error Profiles by Model

Figure 11 shows the distribution of tool-execution error types for four representative models in file reasoning tasks. Different models exhibit substantially different error profiles rather than sharing a single dominant failure mode. For example, `gpt-4.1` is overwhelmingly characterized by `NameError` (over 75% of its errors), which often occurs when later code blocks reuse variables, imports, or intermediate results that are not defined in the current execution context. `Qwen3-32b-rl`, by contrast, is dominated by `SyntaxError`, indicating frequent failures to emit well-formed executable code. `gpt-5` presents the most mixed profile, spreading fairly evenly across `NameError`, `KeyError`/`IndexError`, and environment-related restrictions, while `glm-5` is again `NameError`-leaning but with a broad tail of environment, dependency, and infrastructure errors. These differences suggest that tool-use robustness in file reasoning is not a single capability, but a combination of execution-state consistency, syntactic validity, schema grounding, and stable environment interaction.

Qwen3-32B Base / SFT / RL: Correct Count and Question Count vs Tool Calls by File Type

*Figure 12.* Correct count and question count versus tool call frequency for Qwen3-32B Base, SFT, and RL models across four file types: `xlsx`, `docx`, `pptx`, and `pdf`. The highest correct counts are generally concentrated within a small number of tool calls, indicating that longer tool-use trajectories do not necessarily provide additional benefit.

## F. Implementation Details for Model Access

For smaller open-source models, we observe that directly enforcing the Hermes tool-call format often leads to poor instruction following. We suspect this is because these models may have seen relatively limited agent-style multi-turn interaction data during post-training, making it difficult for them to reliably follow the prescribed tool-use protocol. Therefore, for these smaller open-source models, we adopt a more explicit system-level instruction and manually construct the interaction format to better align model outputs with the required agent workflow.

For API-access models, we use their official APIs under the same sandbox-based tool interface. To remove ambiguity about the file-access protocol, we provide the following common system prompt:

> The user-provided file has already been uploaded into the sandbox current working directory before your first turn. Access it by its exact filename using a relative path, for example `open('filename.ext')` or `pandas.read_excel('filename.xlsx')`. Do NOT ask the user to upload the file again. Do NOT search `/root`, `/mnt/data`, or any absolute path. First assume the file is present in the current working directory and inspect it there.

This prompt does not reveal any task-specific content or answer information. It only specifies the operational contract between the model and the sandbox: the file is already present, should be accessed by its exact relative filename, and should not be treated as a missing upload or an external web resource. We include this prompt to avoid confounding the evaluation with failures caused by uncertainty about whether the file is already available, where it is mounted, or whether it should be downloaded from the web.

## G. Can Models Really Solve These Tasks?

A natural question is whether the benchmark tasks are genuinely solvable by models through code and terminal tools. In practice, most of the file types in our benchmark like PDF, DOCX, XLSX, and PPTX are inherently machine-readable. With the vast Python ecosystem, libraries such as `pdfplumber`, `python-docx`, `openpyxl`, and `python-pptx`,

along with command-line utilities like `pdftotext` and `unoconv`, allow structured extraction of text, tables, figures, and metadata. These toolchains make it possible for models to perform deterministic reasoning over real documents rather than relying on surface-level pattern matching.

From our experiments, we observed that models often fail not because the data are unextractable, but because they prematurely assume that extraction is infeasible. Typical failure cases occur when the model prematurely stops after a failed keyword search or assumes that a document is image-based when it is actually parsable. When the model instead explores code-based parsing, most questions become straightforward. This suggests that the true bottleneck lies not in accessibility, but in the model's willingness to persist and to apply systematic exploration.

The following code snippet illustrates how the task from the *National Energy Potential Report (PPTX)* can be solved programmatically, extracting state-wise energy potential values directly from embedded XML data inside a PowerPoint file:

```python
import zipfile, xml.etree.ElementTree as ET

NS = {"c": "http://schemas.openxmlformats.org/drawingml/2006/chart"}

def parse_chart_pairs(xml_bytes):
    """Parse (category, value) pairs from a PowerPoint chart XML."""
    root = ET.fromstring(xml_bytes)
    pairs = []
    for ser in root.findall(".//c:ser", NS):
        # read categories (e.g., state abbreviations)
        cats = [pt.findtext("c:v", namespaces=NS).strip()
                for pt in ser.findall(".//c:cat//c:strCache//c:pt", NS)]
        # read numeric values (GW)
        vals = [float(pt.findtext("c:v", namespaces=NS))
                for pt in ser.findall(".//c:val//c:numCache//c:pt", NS)]
        pairs.extend(zip(cats, vals))
    return pairs

def find_max_from_pptx(pptx_path):
    """Return the state and value with the highest potential (GW)."""
    with zipfile.ZipFile(pptx_path, "r") as z:
        chart_files = [n for n in z.namelist()
                if n.startswith("ppt/charts/") and n.endswith(".xml")]
        best = None
        for cf in chart_files:
            for state, val in parse_chart_pairs(z.read(cf)):
                if best is None or val > best[1]:
                    best = (state, val)
        return best

# Example usage:
state, value = find_max_from_pptx("ppt100.pptx")
print(f"Highest potential: {state} = {value:.2f} GW")
```

This simple script illustrates that even data embedded in complex PowerPoint charts are fully accessible to code-based reasoning. For brevity, we only present the key part of the solution here. Of course, the model does not always produce such correct code in a single attempt. In our setup, we allow multiple rounds of code generation and execution, enabling the model to iteratively refine its approach until the parsing and reasoning succeed. This iterative process is essential for bridging the gap between conceptual reasoning and practical tool use, ensuring that the final answers genuinely result from executable workflows rather than manual inference.

## H. Reward Design and LLM-based Evaluation

In our reinforcement learning setup, rewards are assigned based on the correctness of the final answer produced by the agent. We adopt an *LLM-as-a-Judge* approach to evaluate final model outputs in a flexible and scalable manner. For each completed rollout, we compare the model response against a reference standard answer using a separate evaluation model. Specifically, we use **GPT-5-Mini** as the judge model due to its strong reasoning ability and low inference cost. The judge produces

a binary decision indicating whether the model response is correct or incorrect. This decision is then mapped to a scalar reward used for reinforcement learning, as described in Section 4.3: a reward of $+1$ is assigned for correct answers, $0$ for incorrect but structurally valid responses, and $-0.2$ for invalid formatting. Here we show the prompt used in the RL process.

```
You are a judge who needs to determine whether the given answer is correct.

Question: {question}

Standard Answer: {standard_answer}

Model Response: {model_response}

Please carefully analyze whether the model response is consistent with the standard
↪  answer. Consider the following factors:
1. Accuracy of the answer - whether the model response is factually correct
2. Completeness of the answer - whether the model response contains all key
↪  information from the standard answer
3. Precision of the answer - for numerical answers, whether they match exactly; for
↪  text answers, whether they express the same meaning
4. Due to the possibility of carry-over issues, even if the last digit after the
↪  decimal point does not match, the calculation can still be correct.

Please only answer "Correct" or "Incorrect", do not provide any other explanation.

Your judgment:
```

While LLM-based evaluation may introduce a degree of stochasticity, we find it to be a practical and effective solution for judging complex file reasoning outputs that involve execution traces and intermediate computations. We leave the exploration of more deterministic or hybrid evaluation strategies to future work.

## I. Qualitative Analysis of Agent Trajectories

To better understand the behavioral shifts induced by reinforcement learning, we present two representative case studies comparing FIRE models before and after reinforcement learning. These cases highlight two distinct dimensions of improvement: **(1) Efficiency**, where the agent resolves file-structure obstacles with fewer interaction rounds; and **(2) Targeted Tool Use**, where the agent applies programmatic parsing to extract and filter task-relevant evidence more reliably.

### I.1. Case Study 1: Efficiency Gains in Spreadsheet Structure Handling

**Task ID:** `ce2f2cab-745a...`
**File Type:** XLSX (`item86.xlsx`)
**Question:** *What is the highest pick number in round 6?*

Table 4 contrasts the execution paths. The key difficulty is that the round-number column uses merged cells. A flat reading of the visible cell values makes it appear that only the first listed row belongs to round 6, leading to the incorrect answer 172. The RL model more directly inspects the spreadsheet structure, propagates the merged-cell value across the covered rows, and identifies all round-6 picks before returning the maximum value 175.

### I.2. Case Study 2: Targeted Evidence Extraction in Financial PDFs

**Task ID:** `f031e06b-3d41...`
**File Type:** PDF (`sanxiang_technology_2025_q1.pdf`)
**Question:** *Subtract the total outstanding balance of all external guarantees ending in 2025 from the total book value of assets explicitly described as mortgaged for bank loans.*

Table 5 highlights a shift toward more targeted tool use. The task requires extracting two separate pieces of evidence from the financial report: the book value of assets mortgaged for bank loans and the outstanding balances of external guarantees whose end dates fall in 2025. The SFT model repeatedly searches and recalculates partial evidence, but misses one qualifying guarantee item and returns an incorrect value. The RL model uses programmatic PDF parsing to collect the relevant lines

| Qwen2.5-32B-FIRE (SFT) Trajectory (Incorrect: 172) | | |
|---|---|---|
| **Step** | **Tool** | **Action / Content** |
| 1–3 | `ExecuteCode` | [Partial] Opens the workbook and extracts rows from the visible table. |
| 4–10 | `ExecuteCode` / `ExecuteShell` | [Inefficient] Repeatedly inspects nearby rows and columns but treats the sheet as a flat table. |
| 11–13 | `ExecuteCode` | [Missed Structure] Identifies only the row whose round label is explicitly visible. |
| 14 | `Answer` | [Incorrect] Returns 172, missing later rows whose round label is inherited from the merged cell. |
| **Qwen2.5-32B-FIRE (RL) Trajectory (Correct: 175)** | | |
| **Step** | **Tool** | **Action / Content** |
| 1 | `ExecuteCode` | [Correct] Loads the workbook and inspects the relevant sheet. |
| 2 | `ExecuteCode` | [Structure-Aware] Handles merged cells in the `Rd #` column by propagating the top-left value across the covered rows. |
| 3 | `Answer` | [Correct] Finds all round-6 entries and returns the maximum pick number, 175. |

*Table 4.* Efficiency comparison in spreadsheet reasoning. The RL model solves the task with fewer tool calls by preserving merged-cell structure, while the SFT model repeatedly inspects the sheet as a flat table and misses rows covered by the same merged round label.

more directly, applies the date condition, and computes the correct result.

*Table 5.* Tool-use comparison in financial PDF reasoning. The RL model uses programmatic extraction to gather and filter task-relevant evidence, while the SFT model repeatedly inspects partial text spans and misses one qualifying guarantee item.

| Qwen3-32B-FIRE (SFT) Trajectory (Incorrect: 92,809,196.81) | | |
|---|---|---|
| **Step** | **Tool** | **Action / Content** |
| 1–3 | `ExecuteCode` | [Partial] Loads the PDF and extracts broad text spans from the report. |
| 4–8 | `ExecuteCode` | [Inefficient] Searches for sections related to guarantees and mortgaged assets. |
| 9–12 | `ExecuteCode` | [Incomplete Filtering] Correctly identifies the mortgaged-asset book value 127,809,196.81, but only aggregates 35,000,000 in qualifying guarantee balances. |
| 13–14 | `Answer` | [Incorrect] Returns $127,809,196.81 - 35,000,000 = 92,809,196.81$. |
| **Qwen3-32B-FIRE (RL) Trajectory (Correct: 72,809,196.81)** | | |
| **Step** | **Tool** | **Action / Content** |
| 1 | `ExecuteCode` | [Targeted] Extracts the PDF text and directly searches for the relevant asset and guarantee sections. |
| 2 | `ExecuteCode` | [Correct] Identifies the mortgaged-asset book value $18,429,667.39 + 109,379,529.42 = 127,809,196.81$. |
| 3 | `ExecuteCode` | [Correct Filtering] Applies the 2025 end-date filter to the external guarantees and aggregates $15,000,000 + 10,000,000 + 10,000,000 + 20,000,000 = 55,000,000$. |
| 4 | `Answer` | [Correct] Returns $127,809,196.81 - 55,000,000 = 72,809,196.81$. |

## J. Structural Complexity in File Reasoning

The following cases highlight a common challenge across file reasoning tasks: the difficulty is often rooted in the internal structure of the file rather than in the surface content alone. In these examples, the answer is not recoverable from a single flat text span or a single cell lookup. Instead, the model must first reconstruct file-specific structure, such as hierarchical spreadsheet headers, temporally anchored quantities in meeting minutes, or embedded objects inside presentation packages. The compared failures therefore reflect breakdowns in structural grounding rather than simple arithmetic mistakes.

### J.1. Case Study 1: Hierarchical Spreadsheet Structure

**File Type:** XLSX (`food-price-outlook_PPIHistoricalForecast.xlsx`)
**Question:** *Using Year of forecast 2011 records in this sample: take the Farm-level milk Upper bound forecast percent change for September ... subtract the Wholesale beef Lower bound forecast percent change for October ... then add the*

| GLM-5 Trajectory (Correct: 63.7) | | |
| --- | --- | --- |
| **Step** | **Tool** | **Action / Content** |
| 1–3 | `ExecuteCode` | [Correct] Opens the workbook, inspects the 2011 and 2012 sheets with `header=None`, and reconstructs both the month row and the `Lower/Mid/Upper` sub-header row. |
| 4–6 | `ExecuteCode` | [Correct] Verifies the target rows and extracts the correct bound-specific values 35.6, 8.9, and 37.0. |
| 7 | `ExecuteCode` | [Correct] Computes $35.6 - 8.9 + 37.0 = 63.7$. |

| GPT-5 Trajectory (Incorrect: 65.0) | | |
| --- | --- | --- |
| **Step** | **Tool** | **Action / Content** |
| 1–4 | `ExecuteShell` / `ExecuteCode` | [Partially Correct] Locates the correct workbook sheets and the relevant month regions. |
| 5–6 | `ExecuteCode` | [Incorrect] Correctly extracts the Farm-level milk September upper-bound value 35.6 and Farm-level wheat February midpoint value 37.0, but misbinds the Wholesale beef October lower-bound column. |
| 7 | `ExecuteCode` | [Incorrect] Extracts 7.6 for Wholesale beef October instead of the correct lower-bound value 8.9. |
| 8 | `ExecuteCode` | [Incorrect] Computes $35.6 - 7.6 + 37.0 = 65.0$, rather than $35.6 - 8.9 + 37.0 = 63.7$. |

*Table 6.* Hierarchical spreadsheet structure. GPT-5 reaches the correct month region but fails to preserve alignment with the second-level `Lower/Mid/Upper` header structure for one target column.

*Farm-level wheat Mid point forecast percent change for February ...*

**Analysis.**   Table 6 summarizes the two trajectories. This case is structurally difficult because the required values are organized under a hierarchical header rather than a flat table. The month label identifies only the top-level column group, while the semantic distinction required by the question is carried by a second header layer: `Lower`, `Mid`, and `Upper`. GLM-5 reconstructs this two-level structure and preserves the alignment through the final extraction step. GPT-5 reaches the correct month regions and recovers two of the three required values, but the second-level header alignment drifts for the Wholesale beef October entry. As a result, it extracts a nearby value from the correct month group but the wrong bound-specific sub-column. The error is therefore tightly linked to spreadsheet structure: solving the task requires maintaining the relationship between grouped month headers and their bound-specific sub-columns.

### J.2. Case Study 2: Temporal Structure in Meeting Minutes

**File Type:** DOCX (`meeting_report.docx`)
**Question:** *Calculate the sum of two values: 1) the patient list size at the time of the last meeting, and 2) the average wasted time in minutes per individual GP "Did Not Attend" (DNA) appointment across the period January to March ...*

**Analysis.**   Table 7 compares how the two models handle the temporal relationship in the meeting minutes. The difficulty in this case comes from the structure of the meeting minutes rather than from document accessibility. The required quantity is not stated explicitly as a standalone value. Instead, it must be reconstructed from a temporally structured update: the document gives the current patient list size and the change since the last meeting, while the question asks for the value at the earlier time point. GLM-5 follows this document structure and recovers the historical quantity before performing the final calculation. GPT-5 reads the same section successfully, but grounds the answer to the current value instead of the temporally referenced one. This shows that even plain-text `docx` files can contain non-trivial structure, where solving the task depends on preserving relationships across time rather than extracting a single visible number.

### J.3. Case Study 3: Multi-Layer Object Structure in Presentations

**File Type:** PPTX (`ppt101.pptx`)
**Question:** *What is the result of the following calculation: Start with the number of years David Barlex taught in comprehensive schools. Subtract the number of individuals credited as authors for Chapter 9. From this intermediate result ...*

| | | GLM-5 Trajectory (Correct: 7959) |
|---|---|---|
| **Step** | **Tool** | **Action / Content** |
| 1 | ExecuteShell | [Correct] Directly loads the local docx file using python-docx. |
| 2–3 | ExecuteCode | [Correct] Extracts the relevant meeting-minute text and identifies both the current patient list size and the reported decrease since the last meeting. |
| 4 | ExecuteCode | [Correct] Reconstructs the required historical value: patient list size at the last meeting = $7917 + 32 = 7949$. |
| 5 | ExecuteCode | [Correct] Computes the average GP DNA wasted time: total wasted time 810 minutes over 81 GP DNA appointments, giving 10. |
| 6 | ExecuteCode | [Correct] Returns $7949 + 10 = 7959$. |
| | | GPT-5 Trajectory (Incorrect: 7927) |
| **Step** | **Tool** | **Action / Content** |
| 1 | ExecuteShell | [Correct] Confirms that the local file exists. |
| 2 | ExecuteShell | [Correct] Successfully reads the document content using python-docx. |
| 3 | ExecuteCode | [Incorrect] Produces a final numeric answer without reconstructing the list size *at the time of the last meeting* from the reported change. |
| 4 | ExecuteCode | [Incorrect] Returns 7927, effectively grounding to the current list size rather than the historical one required by the question. |

*Table 7.* Temporal structure in meeting minutes. Both models read the same local document, but only GLM-5 preserves the temporal relationship needed to reconstruct the historical quantity.

| | | GLM-5 Trajectory (Correct: 18) |
|---|---|---|
| **Step** | **Tool** | **Action / Content** |
| 1–2 | ExecuteCode | [Correct] Loads the presentation and extracts text from all slides. |
| 3–4 | ExecuteShell | [Correct] Unzips the pptx archive and inspects slide relationships together with embedded objects. |
| 5–6 | ExecuteCode | [Correct] Opens the embedded Word documents inside the presentation and extracts the technology-subject table needed for the calculation. |
| 7 | ExecuteCode | [Correct] Combines the years-teaching value, author count, and embedded table evidence to compute 18. |
| | | Gemini-2.5-Pro Trajectory (Incorrect: 9) |
| **Step** | **Tool** | **Action / Content** |
| 1 | ExecuteCode | [Execution Error] Attempts to install and use python-pptx inside a Python block, producing a syntax error. |
| 2 | ExecuteShell | [Recovery] Installs or verifies python-pptx through the shell. |
| 3 | ExecuteCode | [Partially Correct] Extracts visible slide text using python-pptx and identifies the years-teaching value and Chapter 9 author count. |
| 4 | Answer | [Missed Embedded Object] Does not unpack the pptx package or inspect the embedded Word documents, miscounts the computer-science-exclusive technologies as 1 instead of 2, and returns 9. |

*Table 8.* Multi-layer object structure in PPTX. GLM-5 follows the internal object structure of the presentation package, whereas Gemini-2.5-Pro remains at the visible slide-text layer and misses evidence stored in embedded Word documents.

**Analysis.** Table 8 shows how the two models differ in traversing the internal object structure of the presentation package. This case is difficult because the required evidence is distributed across multiple structural layers of the presentation package. The visible slide text provides only part of the information, while the technology-subject mapping needed for the final calculation is stored in embedded Word documents linked from the `pptx` archive. GLM-5 treats the presentation as a structured container: it unpacks the archive, follows internal relationships, and retrieves evidence from the embedded objects before answering. Gemini-2.5-Pro reads the visible slide text and recovers some surface-level evidence, including the years-teaching value and the Chapter 9 author count, but it does not traverse into the embedded Word documents. As a result, it miscounts the number of technologies taught exclusively in computer science and returns 9 instead of 18. This example shows that some `pptx` tasks require package-level reasoning rather than slide-level text extraction alone.

### J.4. Case Study 4: Layout-Preserving Parsing in PDFs

**File Type:** PDF (`Facilities%20Time%20Data%202021%202022.pdf`)
**Question:** *Calculate the ratio of the total FTE of employees undertaking trade union duties to the number of relevant union officials who spent a non-zero percentage but less than 100% of their working hours on facility time ...*

Table 9 shows how an apparent downstream binding error can be caused by an earlier parsing choice. The file is accessible in both trajectories, but the models differ in whether the extraction method preserves the visual table layout needed to identify the correct denominator.

| GLM-5 Trajectory (Correct: 0.978) | | |
|---|---|---|
| **Step** | **Tool** | **Action / Content** |
| 1 | ExecuteShell | [Correct] Uses `pdftotext -layout` to preserve the visual table structure of the PDF. |
| 2 | ExecuteCode | [Correct] Extracts the total FTE value (12.72) and the correct count of officials with non-zero but less than 100% facility time (13). |
| 3 | ExecuteCode | [Correct] Computes $12.72/13 = 0.978$. |

| GPT-4.1 Trajectory (Incorrect: 12.720) | | |
|---|---|---|
| **Step** | **Tool** | **Action / Content** |
| 1 | ExecuteCode | [Partially Correct] Attempts extraction with `PyPDF2`. |
| 2 | ExecuteCode | [Partially Correct] Switches to `pdfminer`, but the extracted text includes encoding noise such as `(cid:14)` and degraded table structure. |
| 3 | ExecuteCode | [Incorrect] Parses the corrupted text as if the table indicated 0 officials in the 1–50% band and 1 official in the 51–99% band. |
| 4 | ExecuteCode | [Incorrect] Uses the correct FTE value (12.72) but divides by the wrong official count (1), yielding 12.720. |

*Table 9.* Layout-preserving parsing in PDF. GLM-5 uses a layout-aware extraction pipeline, while GPT-4.1 relies on generic PDF text extraction that degrades the table structure and leads to an incorrect denominator.

**Analysis.** This case depends on recovering a small numerical table from a PDF rather than merely extracting prose. GLM-5 uses a layout-preserving conversion method, which keeps the row and column relationships intact and makes the subsequent calculation straightforward. GPT-4.1 instead relies on generic PDF text extraction, and the resulting text stream contains both encoding artifacts and weakened table boundaries. The final error appears as a binding mistake in the denominator, but that mistake follows directly from the earlier loss of structural information. For table-heavy `pdf` files, the parsing method itself can determine whether later numerical grounding remains reliable.

### J.5. Case Study 5: Hidden Grouping Structure in XLSX

**File Type:** XLSX (`rice-yearbook_Export-prices-Thailand-Vietnam-India-and-Pakistan.xlsx`)
**Question:** *In 2023, compute: (Thailand A.1 SUPER f.o.b. price for the earliest available month in 2023) + (Vietnam JASMINE (AROMATIC) price in August 2023 minus its price in March 2023) - (Pakistan 25-PERCENT BROKEN price in November 2023) ...*

| | | GLM-5 Trajectory (Correct: 150) |
|---|---|---|
| **Step** | **Tool** | **Action / Content** |
| 1 | ExecuteCode | [Correct] Lists workbook sheets and identifies the relevant tables: Table 25, Table 26, and Table 28. |
| 2–4 | ExecuteCode | [Correct] Reads each sheet with header=None and inspects the raw layout instead of assuming a flat header structure. |
| 5 | ExecuteCode | [Correct] Recovers the structural pattern of the workbook: the first column contains marketing-year block labels (e.g., 2022/23:, 2023/24:), while the second column contains calendar months. |
| 6 | ExecuteCode | [Correct] Correctly determines that for Thailand A.1 SUPER, the earliest *available* month in calendar year 2023 is January 2023, with value 472, since August–December 2022 in the 2022/23 block are marked NQ. |
| 7 | ExecuteCode | [Correct] Extracts Vietnam JASMINE values 758 (August 2023) and 545 (March 2023), and Pakistan 25-PERCENT BROKEN value 535 (November 2023). |
| 8 | ExecuteCode | [Correct] Computes $472 + (758 - 545) - 535 = 150$. |
| | | **Qwen2.5-72B Trajectory (Failed: max_iter_exceeded)** |
| **Step** | **Tool** | **Action / Content** |
| 1–3 | ExecuteCode / ExecuteShell | [Partially Correct] Confirms that the workbook exists and eventually identifies the relevant sheets. |
| 4–10 | ExecuteCode | [Unstable] Repeatedly reloads sheets, tests different reading patterns, and inspects sheet contents without stabilizing a consistent schema. |
| 11–14 | ExecuteCode | [Incorrect] Extracts the year from the first column and filters rows where that extracted year equals 2023, thereby selecting the marketing-year grouping row 2023/24: rather than an actual month row. |
| 15 | ExecuteCode | [Incorrect] Also derives the month from the first column, so the grouping row is misinterpreted as having Month = 2023. |
| 16 | ExecuteCode | [Incorrect] Concludes that the Thailand A.1 SUPER value for the earliest month in 2023 is NaN, keeps probing the workbook structure, and terminates with max_iter_exceeded. |

*Table 10.* Hidden grouping structure in XLSX. GLM-5 reconstructs the distinction between marketing-year group headers and month rows, whereas Qwen2.5-72B treats a grouping row as if it were a data row and becomes trapped in repeated structural probing.

**Analysis.** Table 10 compares a trajectory that reconstructs the hidden grouping structure with one that mistakes a grouping row for a data row. The first column does not directly encode calendar-year observations; instead, it contains marketing-year block labels such as 2022/23: and 2023/24:, while the actual month values appear in the second column. GLM-5 reconstructs this organization explicitly and then resolves the query against the correct calendar-month rows. Qwen2.5-72B does not preserve that distinction. It extracts 2023 from the grouping row 2023/24:, treats that row as if it were a month-level observation, and then interprets the resulting NaN as missing data rather than as evidence of a structural misread. The failure is therefore not a simple arithmetic mistake. It arises because the model confuses structural separators with data records and never recovers from that initial misinterpretation.

## K. Full Tool-Use Trace for Multi-Sheet Excel Reasoning

Here we provide an example of a complete reasoning trajectory generated by the model. This example corresponds to a question involving multi-sheet Excel processing. It illustrates how the model explores the file structure, identifies the relevant headers, corrects intermediate errors, and retrieves values across different demographic sheets. The trajectory captures the full sequence of operations, including initial missteps, inspections of partially relevant rows, adjustments to search strategies, and the final successful extraction of all required values. By showing this end-to-end process, we aim to give readers a clearer understanding of how tool-augmented reasoning unfolds in practice when handling complex spreadsheet inputs.

The model began by examining the overall structure of the Excel file. It first called pd.ExcelFile to obtain the list of sheet names, confirming that the dataset was organized by demographic groups. The model then attempted to load the first sheet using xls.sheet_names[0], but the variable xls had not been defined in that execution context. This resulted in an initial failure with a NameError. The model corrected the issue by reintroducing xls = pd.ExcelFile(file_path) and successfully re-ran the code.

Upon printing the top rows of the sheet, the model discovered that the table used a two-row header structure: the first row contained the year ranges (such as 1977–1978 and 2003–2004), and the second row contained corresponding Mean and SE of mean entries. From this layout, the model inferred that the 2003–2004 Mean value consistently appeared in column index 9.

Next, the model proceeded to read the relevant sheets: Race – Non-Hispanic Black, Race – Hispanic, and Ages 65 and above. In each sheet, it identified the true header row by locating the row containing the text "Food group," and then located the 2003–2004 column by examining the first row. For the Non-Hispanic Black sheet, the model successfully matched the row where the food group was "Energy" and the food source was "FAFH: Restaurant," obtaining the value 138.7.

The Hispanic sheet required additional effort. The model initially attempted a direct conditional match for "Discretionary fats" and "FAFH: Restaurant," but this failed to return any result. It then attempted to print the variable `value`, which had not yet been assigned, resulting in another `NameError`. This erroneous attempt also printed a long metadata row unintentionally. Recognizing the mismatch, the model printed rows 20–35 for inspection and identified the target at row 27, labeled "Discretionary fats – FAFH: Restaurant." Using this row index and the known Mean column, the model retrieved the value 21.45.

For the Ages 65 and above sheet, the model performed a broader search for any row containing the text "Grains, total." This search identified several related rows, including "Grains, total – FAFH: Restaurant" at row 160. Reading the value at the 2003–2004 Mean column yielded 3.06.

With the three required values collected (138.7, 21.45, and 3.06), the model computed the final expression:

$$138.7 - 21.45 - 3.06 = 114.19.$$

This example illustrates a complete tool-use reasoning process that involves structural inspection of a complex dataset, correction of earlier execution errors, row-level inspection when exact string matching fails, and targeted extraction of values across multiple sheets.

---

**Full Reasoning Trace**

```
<think>
First, the user wants me to compute a specific value using 2003-2004-Mean values from an Excel file. The
↪  file is named "food-consumption-nutrient.xlsx".

The computation is:

Energy from FAFH: Restaurant for the Non-Hispanic Black population
Minus Discretionary fats from FAFH: Restaurant for the Hispanic population
Minus Grains, total from FAFH: Restaurant for adults aged 65 and above

I need to extract these values from the Excel file. Since I'm in a sandbox environment, I'll need to use
↪  Python to read the Excel file. I can use the pandas library for this, which is commonly available in
↪  such environments.

Steps:

Load the Excel file using pandas.
Identify the relevant sheets or data. The file might have multiple sheets, so I need to check which one
↪  contains the data for "Table 7".
Find the rows and columns that correspond to the specified values:
"Energy from FAFH: Restaurant" for "Non-Hispanic Black" population
"Discretionary fats from FAFH: Restaurant" for "Hispanic" population
"Grains, total from FAFH: Restaurant" for "adults aged 65 and above"

The user mentioned "2003-2004-Mean values", so I should look for data from that time period.

I'll write Python code to:

Read the Excel file.
Inspect the structure to understand how the data is organized.
Extract the specific values based on the descriptions.

Since the file name includes "Table-7", it might be that Table 7 is the relevant table. I should look for a
↪  sheet named "Table 7" or similar.

Let me assume the file is in the current directory. I'll use pandas to read it.

Code plan:
```

```
Import pandas.
Read the Excel file, possibly specifying the sheet name if known.
If multiple sheets, I might need to explore which one has the data.
Look for columns or rows that match the descriptions.

The values are:

"Energy from FAFH: Restaurant" { probably a nutrient or food group.
"Non-Hispanic Black" { a population group.
"Discretionary fats from FAFH: Restaurant" { another nutrient.
"Hispanic" { population group.
"Grains, total from FAFH: Restaurant" { food group.
"adults aged 65 and above" { age group.

"FAFH" likely stands for "Food Away From Home", and "Restaurant" is the source.

I need to find where these intersect in the table.

Since it's a complex table, I might need to look for a row with "Energy" and column for "FAFH: Restaurant"
↪  and then filter by population.

Similarly for others.

Let me write code to load the file and inspect it.
</think>
<tool_call>
{"name": "ExecuteCode", "arguments": {"code": "import pandas as pd\n\n# Load the Excel file\nfile_path =
↪  'food-consumption-nutrient.xlsx'\n\n# First, check the sheet names to understand the structure\nxls =
↪  pd.ExcelFile(file_path)\nprint(\"Sheet names:\", xls.sheet_names)\n# Load the first sheet to see the
↪  data structure\ndf = pd.read_excel(file_path, sheet_name=xls.sheet_names[0])\nprint(\"First few rows of
↪  the first sheet:\")\nprint(df.head())\n\n# Check the columns to understand what data we
↪  have\nprint(\"\\nColumns:\")\nprint(df.columns.tolist())"}}
</tool_call>
<tool_response>
{"code": 0, "stdout": "Sheet names: ['US consumers aged 2 and above', 'Sex – Males', 'Sex – Females', 'Ages
↪  2{19', 'Ages 20{64', 'Ages 65 and above', 'Cohort – Boys age 2{19', 'Cohort – Girls age 2{19', 'Cohort –
↪  Men age 20 and above', 'Cohort – Women age 20 and above', 'Income – low-income households', 'Income –
↪  mid-income households', 'Income – high-income households', 'Race – Non-Hispanic White', 'Race –
↪  Non-Hispanic Black', 'Race – Hispanic', 'Race – Other racial & ethnic ', 'Edu. – Less than high school',
↪  'Edu. – High school degree', 'Edu. – College attended']\nFirst few rows of the first sheet:\n Table
↪  7|Density of food group by food source, U.S. consumers aged 2 and above, 1977{2018 ... Unnamed: 24\n0
↪  NaN ... NaN\n1 Food group ... SE of mean\n2 Energy ... 14.12\n3 Energy ... 22.45\n4 Energy ...
↪  25.02\n\n[5 rows x 25 columns]\n\nColumns:\n['Table 7|Density of food group by food source, U.S.
↪  consumers aged 2 and above, 1977{2018', 'Unnamed: 1', 'Unnamed: 2', 'Unnamed: 3', 'Unnamed: 4', 'Unnamed:
↪  5', 'Unnamed: 6', 'Unnamed: 7', 'Unnamed: 8', 'Unnamed: 9', 'Unnamed: 10', 'Unnamed: 11', 'Unnamed: 12',
↪  'Unnamed: 13', 'Unnamed: 14', 'Unnamed: 15', 'Unnamed: 16', 'Unnamed: 17', 'Unnamed: 18', 'Unnamed: 19',
↪  'Unnamed: 20', 'Unnamed: 21', 'Unnamed: 22', 'Unnamed: 23', 'Unnamed: 24']", "stderr": ""}
</tool_response>

<think>
First, I need to understand the structure of the Excel file.
From the initial exploration, I can see that the data is organized in multiple sheets, each for a different
↪  demographic group.
The user is asking for:
Energy from FAFH: Restaurant for the Non-Hispanic Black population
Discretionary fats from FAFH: Restaurant for the Hispanic population
Grains, total from FAFH: Restaurant for adults aged 65 and above
And we need to use the 2003-2004-Mean values only.

Let me examine the structure more carefully to find where the time periods are specified.
I need to look at the columns to see if they contain year information.
</think>

<tool_call>
{"name": "ExecuteCode",
 "arguments": {
  "code": "import pandas as pd\n\n# Load the Excel file\nfile_path = 'food-consumption-nutrient.xlsx'\n\n#
↪  Load the first sheet to see the full column headers\ndf = pd.read_excel(file_path,
↪  sheet_name=xls.sheet_names[0], header=None)\nprint(\"First sheet with no header – first 15
↪  rows:\")\nprint(df.head(15))\n\n# Check if there are row labels that indicate time
↪  periods\nprint(\"\\nRow 0:\")\nprint(df.iloc[0].tolist())\nprint(\"\\nRow
↪  1:\")\nprint(df.iloc[1].tolist())\nprint(\"\\nRow 2:\")\nprint(df.iloc[2].tolist())"}
 }
}
</tool_call>

user

<tool_response>
```

```
{"code": 1,
 "stdout": "",
 "stderr": "Traceback (most recent call last):\n File \"/root/code.py\", line 7, in <module>\n df =
 ↪  pd.read_excel(file_path, sheet_name=xls.sheet_names[0], header=None)\n  ^^^\nNameError: name 'xls' is
 ↪  not defined"}
</tool_response>

assistant

<think>
We are getting an error because the variable 'xls' is not defined in the current execution context.
Let me re-run the code with proper variable initialization.
</think>

<tool_call>
{"name": "ExecuteCode",
 "arguments": {
  "code": "import pandas as pd\n\n# Load the Excel file\nfile_path = 'food-consumption-nutrient.xlsx'\n\n#
  ↪  First, check the sheet names to understand the structure\nxls = pd.ExcelFile(file_path)\nprint(\"Sheet
  ↪  names:\", xls.sheet_names)\n\n# Load the first sheet to see the full structure with no header\ndf =
  ↪  pd.read_excel(file_path, sheet_name=xls.sheet_names[0], header=None)\nprint(\"First sheet with no
  ↪  header - first 15 rows:\")\nprint(df.head(15))\n\n# Check if there are row labels that indicate time
  ↪  periods\nprint(\"\\nRow 0:\")\nprint(df.iloc[0].tolist())\nprint(\"\\nRow
  ↪  1:\")\nprint(df.iloc[1].tolist())\nprint(\"\\nRow 2:\")\nprint(df.iloc[2].tolist())\nprint(\"\\nRow
  ↪  3:\")\nprint(df.iloc[3].tolist())"}
 }
}
</tool_call>

user

<tool_response>
{"code": 0,
 "stdout": "Sheet names: ['US consumers aged 2 and above', 'Sex - Males', 'Sex - Females', 'Ages 2{19',
 ↪  'Ages 20{64', 'Ages 65 and above', 'Cohort - Boys age 2{19', 'Cohort - Girls age 2{19', 'Cohort - Men
 ↪  age 20 and above', 'Cohort - Women age 20 and above', 'Income - low-income households', 'Income -
 ↪  mid-income households', 'Income - high-income households', 'Race - Non-Hispanic White', 'Race -
 ↪  Non-Hispanic Black', 'Race - Hispanic', 'Race - Other racial & ethnic ', 'Edu. - Less than high school',
 ↪  'Edu. - High school degree', 'Edu. - College attended']\nFirst sheet with no header - first 15 rows:\n
 ↪  0 ... 24\n0 Table 7|Density of food group by food source, ... ... NaN\n1 NaN ... NaN\n2 Food group ...
 ↪  SE of mean\n3 Energy ... 14.12\n4 Energy ... 22.45\n5 Energy ... 25.02\n6 Energy ... 10.87\n7 Energy ...
 ↪  17.92\n8 Energy ... 4.88\n9 Energy ... 6.27\n10 Added sugars ... 0.18\n11 Added sugars ... 0.25\n12
 ↪  Added sugars ... 0.25\n13 Added sugars ... 0.32\n14 Added sugars ... 0.26\n\n[15 rows x 25
 ↪  columns]\n\nRow 0:\n['Table 7|Density of food group by food source, U.S. consumers aged 2 and above,
 ↪  1977{2018', nan, nan, nan, nan, nan, nan, nan, nan, nan, nan, nan, nan, nan, nan, nan, nan, nan, nan,
 ↪  nan, nan, nan, nan, nan, nan]\nRow 1:\n[nan, nan, '1977{1978', nan, '1989{1991', nan,
 ↪  '1994{1998', nan, '2003{2004', nan, '2005{2006', nan, '2007{2008', nan, '2009{2010', nan, '2011{2012',
 ↪  nan, '2013{2014', nan, '2015{2016', nan, '2017{2018', nan]\n\nRow 2:\n['Food group', 'Food source',
 ↪  'Measurement', 'Mean', 'SE of mean', 'Mean', 'SE of mean', 'Mean', 'SE of mean', 'Mean', 'SE of mean',
 ↪  'Mean', 'SE of mean', 'Mean', 'SE of mean', 'Mean', 'SE of mean', 'Mean', 'SE of mean', 'Mean', 'SE of
 ↪  mean', 'Mean', 'SE of mean', 'Mean', 'SE of mean']\n\nRow 3:\n['Energy ', 'Total', 'Calories', 1806.88,
 ↪  12.39, 1847.48, 18.9, 2033.93, 16.2, 2194.78, 15.59, 2157.35, 29.02, 2069.79, 24.55, 2080.56, 12.91,
 ↪  2138.89, 12.34, 2079.32, 17.17, 2048.24, 18.25, 2093.14, 14.12]",
 "stderr": ""}
</tool_response>

assistant

<think>
First, I need to understand the structure of the data.
From the output, I can see that:
Row 1 contains the time periods: 1977{1978, 1989{1991, 1994{1998, 2003{2004, 2005{2006, 2007{2008, 2009{2010,
↪  2011{2012, 2013{2014, 2015{2016, 2017{2018.
Row 2 contains the measurements: Mean, SE of mean, Mean, SE of mean, etc.
The data starts from row 3 onwards.

The user wants values for 2003-2004-Mean, so I need to find the column for 2003-2004 Mean.
From row 1, 2003-2004 is at index 9 (since Python is 0-indexed, but let's check the exact position).

Looking at row 1:
Index 3: '1977{1978
Index 4: nan (but from row 2, it's 'Mean' for 1977-1978, so index 3 is Mean, index 4 is SE of mean)
Similarly, for 2003-2004, it should be at index 9 (Mean) and index 10 (SE of mean)

But let's confirm the column indices.
I should load the data with proper headers.

Since the data is structured with multiple header rows, I need to parse it carefully.
I'll load the data again and set the headers appropriately.
```

```
The user's query involves:
Energy from FAFH: Restaurant for Non-Hispanic Black population
Discretionary fats from FAFH: Restaurant for Hispanic population
Grains, total from FAFH: Restaurant for adults aged 65 and above

I need to find these values for 2003-2004 Mean.

Let me first understand the column structure for the 2003-2004 period.
From row 1, 2003-2004 is at index 9, and from row 2, it's 'Mean' at index 9 and 'SE of mean' at index 10.
So for 2003-2004 Mean, I need column index 9.

Now, I need to find the rows for:
Energy from FAFH: Restaurant for Non-Hispanic Black
Discretionary fats from FAFH: Restaurant for Hispanic
Grains, total from FAFH: Restaurant for ages 65 and above

I should load the specific sheets for each population group.
Let me load the sheets one by one.

First, for Non-Hispanic Black population, I need the sheet 'Race - Non-Hispanic Black'.
Second, for Hispanic population, sheet 'Race - Hispanic'.
Third, for adults aged 65 and above, sheet 'Ages 65 and above'.

I'll load each sheet and find the relevant rows.
Let me start with the Non-Hispanic Black sheet.
</think>

<tool_call>
{"name": "ExecuteCode",
 "arguments": {
  "code": "import pandas as pd\n\n# Load the Excel file\nfile_path = 'food-consumption-nutrient.xlsx'\n\n#
  ↪  Load the Non-Hispanic Black sheet with no header to see the structure\ndf_black =
  ↪  pd.read_excel(file_path, sheet_name='Race - Non-Hispanic Black', header=None)\nprint(\"Non-Hispanic
  ↪  Black sheet - first 10 rows:\")\nprint(df_black.head(10))\n\n# Check the row that contains 'Food
  ↪  group' to identify the header row\nfor i in range(10):\n if 'Food group' in str(df_black.iloc[i,
  ↪  0]):\n  print(f\"Header row found at index {i}: {df_black.iloc[i].tolist()}\")\n  break\n\n# Also
  ↪  check row 1 for time periods\nprint(\"\\nRow 1 of Non-Hispanic Black
  ↪  sheet:\")\nprint(df_black.iloc[1].tolist())"}
 }
}
</tool_call>

user

<tool_response>
{"code": 0,
 "stdout": "Non-Hispanic Black sheet - first 10 rows:\n 0 ... 24\n0 Table 7E2|Density of food group by food
 ↪  source... ... NaN\n1 NaN ... NaN\n2 Food group ... SE of mean\n3 Energy ... 25.98\n4 Energy ...
 ↪  35.76\n5 Energy ... 30.37\n6 Energy ... 16.87\n7 Energy ... 30.77\n8 Energy ... 4.9\n9 Energy ...
 ↪  9.17\n\n[10 rows x 25 columns]\nHeader row found at index 2: ['Food group', 'Food source',
 ↪  'Measurement', 'Mean', 'SE of mean', 'Mean', 'SE of mean', 'Mean', 'SE of mean', 'Mean', 'SE of mean',
 ↪  'Mean', 'SE of mean', 'Mean', 'SE of mean', 'Mean', 'SE of mean', 'Mean', 'SE of mean', 'Mean', 'SE of
 ↪  mean', 'Mean', 'SE of mean', 'Mean', 'SE of mean']\n\nRow 1 of Non-Hispanic Black sheet:\n[nan, nan,
 ↪  nan, '1977{1978', nan, '1989{1991', nan, '1994{1998', nan, '2003{2004', nan, '2005{2006', nan,
 ↪  '2007{2008', nan, '2009{2010', nan, '2011{2012', nan, '2013{2014', nan, '2015{2016', nan, '2017{2018',
 ↪  nan]",
 "stderr": ""}
</tool_response>

assistant

<think>
First, I need to understand the structure of the data.
From the output, I can see that:
- The data has multiple sheets for different demographic groups.
- The header row is at index 2, which contains the column names including time periods.
- Row 1 contains the time periods: 1977{1978, 1989{1991, 1994{1998, 2003{2004, 2005{2006, 2007{2008,
↪  2009{2010, 2011{2012, 2013{2014, 2015{2016, 2017{2018.
- The columns alternate between "Mean" and "SE of mean" for each time period.

I need to find the values for 2003-2004 Mean only.
So for each time period, the "Mean" value is in one column and "SE of mean" in the next.
For 2003-2004, the "Mean" should be at column index where the time period is '2003{2004'.
From row 1, I see that '2003{2004' is at index 9 (since Python is 0-indexed, but let's confirm the exact
↪  position).

Let me map the columns:
Column 3: 1977{1978 Mean
Column 4: 1977{1978 SE of mean
Column 5: 1989{1991 Mean
```

```
Column 6: 1989{1991 SE of mean
Column 7: 1994{1998 Mean
Column 8: 1994{1998 SE of mean
Column 9: 2003{2004 Mean
Column 10: 2003{2004 SE of mean
And so on...

So for 2003-2004 Mean, it should be at column index 9.

Now, I need to find three specific values:
Energy from FAFH: Restaurant for Non-Hispanic Black population
Discretionary fats from FAFH: Restaurant for Hispanic population
Grains, total from FAFH: Restaurant for adults aged 65 and above

I need to load the appropriate sheets:
For Non-Hispanic Black: sheet 'Race - Non-Hispanic Black'
For Hispanic: sheet 'Race - Hispanic'
For adults aged 65 and above: sheet 'Ages 65 and above'

I'll load each sheet and extract the value for 2003-2004 Mean for the specified food group and food source.
Let me start with the Non-Hispanic Black sheet for Energy from FAFH: Restaurant.
</think>

<tool_call>
{"name": "ExecuteCode",
 "arguments": {
  "code": "import pandas as pd\n\n# Load the Excel file\nfile_path = 'food-consumption-nutrient.xlsx'\n\n#
  ↪  Load the Non-Hispanic Black sheet\ndf_black = pd.read_excel(file_path, sheet_name='Race - Non-Hispanic
  ↪  Black', header=None)\n\n# Find the header row (where 'Food group' is in column 0)\nheader_row =
  ↪  None\nfor i in range(len(df_black)):\n if 'Food group' in str(df_black.iloc[i, 0]):\n  header_row =
  ↪  i\n  break\n\nprint(f\"Header row at index: {header_row}\")\nprint(\"Header row
  ↪  values:\")\nprint(df_black.iloc[header_row].tolist())\n\n# The time periods are in row 1 (index
  ↪  1)\nprint(\"\\nTime periods in row 1:\")\nprint(df_black.iloc[1].tolist())\n\n# Find the column index
  ↪  for 2003-2004 Mean\nfor idx, val in enumerate(df_black.iloc[1]):\n if val == '2003{2004':\n
  ↪  print(f\"2003-2004 found at column index {idx}\")\n  mean_col = idx\n  se_col = idx + 1\n
  ↪  print(f\"Mean column: {mean_col}, SE column: {se_col}\")\n  break\n\n# Now, find the row for 'Energy'
  ↪  and 'FAFH: Restaurant'\nfor i in range(header_row + 1, len(df_black)):\n food_group = df_black.iloc[i,
  ↪  0]\n food_source = df_black.iloc[i, 1]\n if food_group == 'Energy ' and food_source == 'FAFH:
  ↪  Restaurant':\n  print(f\"Found Energy from FAFH: Restaurant at row {i}\")\n  value = df_black.iloc[i,
  ↪  mean_col]\n  print(f\"Value: {value}\")\n  break"
 }
}
</tool_call>

user

<tool_response>
{"code": 0,
 "stdout": "Header row at index: 2\nHeader row values:\n['Food group', 'Food source', 'Measurement', 'Mean',
 ↪  'SE of mean', 'Mean', 'SE of mean', 'Mean', 'SE of mean', 'Mean', 'SE of mean', 'Mean', 'SE of mean',
 ↪  'Mean', 'SE of mean', 'Mean', 'SE of mean', 'Mean', 'SE of mean', 'Mean', 'SE of mean', 'Mean', 'SE of
 ↪  mean', 'Mean', 'SE of mean']\n\nTime periods in row 1:\n[nan, nan, nan, '1977{1978', nan, '1989{1991',
 ↪  nan, '1994{1998', nan, '2003{2004', nan, '2005{2006', nan, '2007{2008', nan, '2009{2010', nan,
 ↪  '2011{2012', nan, '2013{2014', nan, '2015{2016', nan, '2017{2018', nan]\n2003-2004 found at column
 ↪  index 9\nMean column: 9, SE column: 10\nFound Energy from FAFH: Restaurant at row 6\nValue: 138.7",
 "stderr": ""}
</tool_response>

assistant

<think>
First, I need to compute the expression:
(Energy from FAFH: Restaurant for Non-Hispanic Black)
minus
(Discretionary fats from FAFH: Restaurant for Hispanic)
minus
(Grains, total from FAFH: Restaurant for adults aged 65 and above).

I have the value for Energy from FAFH: Restaurant for Non-Hispanic Black: 138.7.
Now I need to find the other two values.

Let me find the value for Discretionary fats from FAFH: Restaurant for Hispanic population.
</think>

<tool_call>
{"name": "ExecuteCode",
 "arguments": {
```

```
   "code": "import pandas as pd\n\n# Load the Excel file\nfile_path = 'food-consumption-nutrient.xlsx'\n\n#
↪   Load the Hispanic sheet\ndf_hispanic = pd.read_excel(file_path, sheet_name='Race - Hispanic',
↪   header=None)\n\n# Find the header row\nheader_row = None\nfor i in range(len(df_hispanic)):\n if 'Food
↪   group' in str(df_hispanic.iloc[i, 0]):\n  header_row = i\n  break\n\nprint(f\"Header row at index:
↪   {header_row}\")\nprint(\"Header row values:\")\nprint(df_hispanic.iloc[header_row].tolist())\n\n# Find
↪   the column index for 2003-2004 Mean\nfor idx, val in enumerate(df_hispanic.iloc[1]):\n if val ==
↪   '2003{2004':\n  print(f\"2003-2004 found at column index {idx}\")\n  mean_col = idx\n  se_col = idx +
↪   1\n  print(f\"Mean column: {mean_col}, SE column: {se_col}\")\n  break\n\n# Find the row for
↪   'Discretionary fats' and 'FAFH: Restaurant'\nfor i in range(header_row + 1, len(df_hispanic)):\n
↪   food_group = df_hispanic.iloc[i, 0]\n food_source = df_hispanic.iloc[i, 1]\n if food_group ==
↪   'Discretionary fats' and food_source == 'FAFH: Restaurant':\n  print(f\"Found Discretionary fats from
↪   FAFH: Restaurant at row {i}\")\n  value = df_hispanic.iloc[i, mean_col]\n  print(f\"Value:
↪   {value}\")\n  break"}
  }
}
</tool_call>

user

<tool_response>
{"code": 0,
 "stdout": "Header row at index: 2\nHeader row values:\n['Food group', 'Food source', 'Measurement', 'Mean',
↪   'SE of mean', 'Mean', 'SE of mean', 'Mean', 'SE of mean', 'Mean', 'SE of mean', 'Mean', 'SE of mean',
↪   'Mean', 'SE of mean', 'Mean', 'SE of mean', 'Mean', 'SE of mean', 'Mean', 'SE of mean', 'Mean', 'SE of
↪   mean', 'Mean', 'SE of mean']\n2003-2004 found at column index 9\nMean column: 9, SE column: 10",
 "stderr": ""}
</tool_response>

assistant

<think>
We are looking for 'Discretionary fats' and 'FAFH: Restaurant' in the Hispanic sheet.
Let's check the rows to see if we can find the exact match.
</think>

<tool_call>
{"name": "ExecuteCode",
 "arguments": {
  "code": "import pandas as pd\n\n# Load the Excel file\nfile_path = 'food-consumption-nutrient.xlsx'\n\n#
↪   Load the Hispanic sheet\ndf_hispanic = pd.read_excel(file_path, sheet_name='Race - Hispanic',
↪   header=None)\n\n# Find the header row\nheader_row = None\nfor i in range(len(df_hispanic)):\n if 'Food
↪   group' in str(df_hispanic.iloc[i, 0]):\n  header_row = i\n  break\nprint(f\"Header row at index:
↪   {header_row}\")\n\n# Print the first few rows to see the structure\nprint(\"First 20 rows of Hispanic
↪   sheet:\")\nfor i in range(header_row, header_row + 20):\n if i < len(df_hispanic):\n  print(f\"Row {i}:
↪   {df_hispanic.iloc[i, 0]} - {df_hispanic.iloc[i, 1]}\")\n\n# Let's search for 'Discretionary fats'
↪   specifically\nprint(\"\\nSearching for 'Discretionary fats':\")\nfor i in range(header_row + 1,
↪   len(df_hispanic)):\n food_group = df_hispanic.iloc[i, 0]\n if 'Discretionary fats' in
↪   str(food_group):\n  print(f\"Row {i}: {food_group} - {df_hispanic.iloc[i, 1]}\")"}
 }
}
</tool_call>

user

<tool_response>
{"code": 0,
 "stdout": "Header row at index: 2\nFirst 20 rows of Hispanic sheet:\nRow 2: Food group - Food source\nRow 3:
↪   Energy - Total\nRow 4: Energy - FAH\nRow 5: Energy - FAFH\nRow 6: Energy - FAFH: Restaurant\nRow 7:
↪   Energy - FAFH: Fast food\nRow 8: Energy - FAFH: School\nRow 9: Energy - FAFH: Others\nRow 10: Added
↪   sugars - Total\nRow 11: Added sugars - FAH\nRow 12: Added sugars - FAFH\nRow 13: Added sugars - FAFH:
↪   Restaurant\nRow 14: Added sugars - FAFH: Fast food\nRow 15: Added sugars - FAFH: School\nRow 16: Added
↪   sugars - FAFH: Others\nRow 17: Discretionary fats and oils - Total\nRow 18: Discretionary fats and oils
↪   - FAH\nRow 19: Discretionary fats and oils - FAFH\nRow 20: Discretionary fats and oils - FAFH:
↪   Restaurant\nRow 21: Discretionary fats and oils - FAFH: Fast food\n\nSearching for 'Discretionary
↪   fats':\nRow 17: Discretionary fats and oils - Total\nRow 18: Discretionary fats and oils - FAH\nRow 19:
↪   Discretionary fats and oils - FAFH\nRow 20: Discretionary fats and oils - FAFH: Restaurant\nRow 21:
↪   Discretionary fats and oils - FAFH: Fast food\nRow 22: Discretionary fats and oils - FAFH: School\nRow
↪   23: Discretionary fats and oils - FAFH: Others\nRow 24: Discretionary fats - Total\nRow 25:
↪   Discretionary fats - FAH\nRow 26: Discretionary fats - FAFH\nRow 27: Discretionary fats - FAFH:
↪   Restaurant\nRow 28: Discretionary fats - FAFH: Fast food\nRow 29: Discretionary fats - FAFH:
↪   School\nRow 30: Discretionary fats - FAFH: Others",
 "stderr": ""}
</tool_response>

assistant

<think>
We are looking for 'Discretionary fats' and 'FAFH: Restaurant' in the Hispanic sheet.
From the output, we see that there is a row with 'Discretionary fats' and 'FAFH: Restaurant' at row 27.
Let's get the value for 2003-2004 Mean from that row.
```

```
</think>

<tool_call>
{"name": "ExecuteCode",
 "arguments": {
  "code": "import pandas as pd\n\n# Load the Excel file\nfile_path = 'food-consumption-nutrient.xlsx'\n\n#
  ↪  Load the Hispanic sheet\ndf_hispanic = pd.read_excel(file_path, sheet_name='Race - Hispanic',
  ↪  header=None)\n\n# Find the header row\nheader_row = None\nfor i in range(len(df_hispanic)):\n if 'Food
  ↪  group' in str(df_hispanic.iloc[i, 0]):\n  header_row = i\n  break\n\nprint(f\"Header row at index:
  ↪  {header_row}\")\n\n# Find the column index for 2003-2004 Mean\nfor idx, val in
  ↪  enumerate(df_hispanic.iloc[1]):\n if val == '2003{2004':\n  print(f\"2003-2004 found at column index
  ↪  {idx}\")\n  mean_col = idx\n  se_col = idx + 1\n  print(f\"Mean column: {mean_col}, SE column:
  ↪  {se_col}\")\n  break\n\n# Find the row for 'Discretionary fats' and 'FAFH: Restaurant'\n# From
  ↪  previous output, it should be at row 27\nfor i in range(header_row + 1, len(df_hispanic)):\n
  ↪  food_group = df_hispanic.iloc[i, 0]\n food_source = df_hispanic.iloc[i, 1]\n if food_group ==
  ↪  'Discretionary fats' and food_source == 'FAFH: Restaurant':\n  print(f\"Found Discretionary fats from
  ↪  FAFH: Restaurant at row {i}\")\n  value = df_hispanic.iloc[i, mean_col]\n  print(f\"Value:
  ↪  {value}\")\n  break\n\n# Also, let's confirm the value by printing the row\nprint(f\"\\nRow {i}
  ↪  details:\")\nprint(f\"Food group: {food_group}\")\nprint(f\"Food source:
  ↪  {food_source}\")\nprint(f\"Value at mean_col {mean_col}: {value}\")"}
 }
}
</tool_call>

user

<tool_response>
{"code": 1,
 "stdout": "Header row at index: 2\n2003-2004 found at column index 9\nMean column: 9, SE column: 10\n\nRow
 ↪  257 details:\nFood group: Sources: USDA, Economic Research Service using USDA, Agricultural Research
 ↪  Service, 1977{1978 Nationwide Food Consumption Survey data, and 1989{1991, 1994{1996, and 1998
 ↪  Continuing Survey of Food Intakes by Individuals data; and USDA, Agricultural Research Service and U.S.
 ↪  Department of Health and Human Services, Center for Disease Control and Prevention, 2003{2004,
 ↪  2005{2006, 2007{2008, 2009{2010, 2011{2012, 2013{2014, 2015{2016, and 2017{2018 What We Eat in America,
 ↪  National Health and Nutrition Examination Survey data.\nFood source: nan",
 "stderr": "Traceback (most recent call last):\n File \"/root/code.py\", line 42, in <module>\n
 ↪  print(f\"Value at mean_col {mean_col}: {value}\")\n ^^^^^\nNameError: name 'value' is not defined. Did
 ↪  you mean: 'False'?"}
</tool_response>

assistant

<think>
We are having trouble finding the exact row for 'Discretionary fats' and 'FAFH: Restaurant' in the Hispanic
↪  sheet.
Let's try a different approach:
we know that the row should be around row 27 based on the previous output.
Let's look at the data around row 25 to row 35 to find the exact row.
</think>

<tool_call>
{"name": "ExecuteCode",
 "arguments": {
  "code": "import pandas as pd\n\n# Load the Excel file\nfile_path = 'food-consumption-nutrient.xlsx'\n\n#
  ↪  Load the Hispanic sheet\ndf_hispanic = pd.read_excel(file_path, sheet_name='Race - Hispanic',
  ↪  header=None)\n\n# Find the header row\nheader_row = None\nfor i in range(len(df_hispanic)):\n if 'Food
  ↪  group' in str(df_hispanic.iloc[i, 0]):\n  header_row = i\n  break\n\nprint(f\"Header row at index:
  ↪  {header_row}\")\n\n# Print rows 20 to 35 to find 'Discretionary fats' and 'FAFH: Restaurant'\nfor i in
  ↪  range(20, 36):\n if i < len(df_hispanic):\n  food_group = df_hispanic.iloc[i, 0]\n  food_source =
  ↪  df_hispanic.iloc[i, 1]\n  print(f\"Row {i}: {food_group} - {food_source}\")\n\n# Also, let's check the
  ↪  column indices for 2003-2004 Mean\nfor idx, val in enumerate(df_hispanic.iloc[1]):\n if val ==
  ↪  '2003{2004':\n  print(f\"\\n2003-2004 found at column index {idx}\")\n  mean_col = idx\n  se_col = idx
  ↪  + 1\n  print(f\"Mean column: {mean_col}, SE column: {se_col}\")\n  break"}
 }
}
</tool_call>

user

<tool_response>
{"code": 0,
 "stdout": "Header row at index: 2\nRow 20: Discretionary fats and oils - FAFH: Restaurant\nRow 21:
 ↪  Discretionary fats and oils - FAFH: Fast food\nRow 22: Discretionary fats and oils - FAFH: School\nRow
 ↪  23: Discretionary fats and oils - FAFH: Others\nRow 24: Discretionary fats - Total\nRow 25:
 ↪  Discretionary fats - FAH\nRow 26: Discretionary fats - FAFH\nRow 27: Discretionary fats - FAFH:
 ↪  Restaurant\nRow 28: Discretionary fats - FAFH: Fast food\nRow 29: Discretionary fats - FAFH:
 ↪  School\nRow 30: Discretionary fats - FAFH: Others\nRow 31: Discretionary oils - Total\nRow 32:
 ↪  Discretionary oils - FAH\nRow 33: Discretionary oils - FAFH\nRow 34: Discretionary oils - FAFH:
 ↪  Restaurant\nRow 35: Discretionary oils - FAFH: Fast food\n\n2003-2004 found at column index 9\nMean
 ↪  column: 9, SE column: 10",
```

```
 "stderr": ""}
</tool_response>

assistant

<think>
We are looking for 'Discretionary fats' and 'FAFH: Restaurant' in the Hispanic sheet.
From the output, we see that row 27 has:
Food group: 'Discretionary fats'
Food source: 'FAFH: Restaurant'.
Let's get the value for 2003-2004 Mean from that row.
</think>

<tool_call>
{"name": "ExecuteCode",
 "arguments": {
  "code": "import pandas as pd\n\n# Load the Excel file\nfile_path = 'food-consumption-nutrient.xlsx'\n\n#
  ↪  Load the Hispanic sheet\ndf_hispanic = pd.read_excel(file_path, sheet_name='Race - Hispanic',
  ↪  header=None)\n\n# Find the header row\nheader_row = None\nfor i in range(len(df_hispanic)):\n if 'Food
  ↪  group' in str(df_hispanic.iloc[i, 0]):\n   header_row = i\n   break\n\nprint(f\"Header row at index:
  ↪  {header_row}\")\n\n# Find the column index for 2003-2004 Mean\nfor idx, val in
  ↪  enumerate(df_hispanic.iloc[1]):\n if val == '2003{2004':\n  print(f\"2003-2004 found at column index
  ↪  {idx}\")\n  mean_col = idx\n  se_col = idx + 1\n  print(f\"Mean column: {mean_col}, SE column:
  ↪  {se_col}\")\n  break\n\n# Get the value for Discretionary fats from FAFH: Restaurant at row
  ↪  27\nfood_group = df_hispanic.iloc[27, 0]\nfood_source = df_hispanic.iloc[27, 1]\nvalue =
  ↪  df_hispanic.iloc[27, mean_col]\nprint(f\"\\nRow 27: Food group = '{food_group}', Food source =
  ↪  '{food_source}'\")\nprint(f\"Value for 2003-2004 Mean: {value}\")\n\n# Let's also print the entire row
  ↪  to verify\nprint(f\"\\nFull row 27:\")\nprint(df_hispanic.iloc[27].tolist())"
 }
}
</tool_call>

user

<tool_response>
{"code": 0,
 "stdout": "Header row at index: 2\n2003-2004 found at column index 9\nMean column: 9, SE column: 10\n\nRow
 ↪  27: Food group = 'Discretionary fats ', Food source = 'FAFH: Restaurant'\nValue for 2003-2004 Mean:
 ↪  21.45\n\nFull row 27:\n['Discretionary fats ', 'FAFH: Restaurant', 'Grams per 1,000 calories', 28.34,
 ↪  1.82, 23.92, 0.5, 25.02, 1.34, 21.45, 1.53, 17.07, 0.81, 16.23, 0.83, 18.39, 0.57, 15.98, 0.77, 14.84,
 ↪  1.02, 14.23, 0.78, 15.17, 0.61]",
 "stderr": ""}
</tool_response>

assistant

<think>
We have found the value for Discretionary fats from FAFH: Restaurant for Hispanic population: 21.45.
Now we need to find the value for Grains, total from FAFH: Restaurant for adults aged 65 and above.

Let's load the 'Ages 65 and above' sheet and find the value.
</think>

<tool_call>
{"name": "ExecuteCode",
 "arguments": {
  "code": "import pandas as pd\n\n# Load the Excel file\nfile_path = 'food-consumption-nutrient.xlsx'\n\n#
  ↪  Load the 'Ages 65 and above' sheet\ndf_65plus = pd.read_excel(file_path, sheet_name='Ages 65 and
  ↪  above', header=None)\n\n# Find the header row\nheader_row = None\nfor i in range(len(df_65plus)):\n if
  ↪  'Food group' in str(df_65plus.iloc[i, 0]):\n   header_row = i\n   break\n\nprint(f\"Header row at index:
  ↪  {header_row}\")\n\n# Find the column index for 2003-2004 Mean\nfor idx, val in
  ↪  enumerate(df_65plus.iloc[1]):\n if val == '2003{2004':\n  print(f\"2003-2004 found at column index
  ↪  {idx}\")\n  mean_col = idx\n  se_col = idx + 1\n  print(f\"Mean column: {mean_col}, SE column:
  ↪  {se_col}\")\n  break\n\n# Now, find the row for 'Grains, total' and 'FAFH: Restaurant'\nfor i in
  ↪  range(header_row + 1, len(df_65plus)):\n food_group = df_65plus.iloc[i, 0]\n food_source =
  ↪  df_65plus.iloc[i, 1]\n if food_group == 'Grains, total' and food_source == 'FAFH: Restaurant':\n
  ↪  print(f\"Found Grains, total from FAFH: Restaurant at row {i}\")\n   value = df_65plus.iloc[i,
  ↪  mean_col]\n   print(f\"Value: {value}\")\n   break\n\n# If not found, let's print the rows to see what's
  ↪  available\nif 'value' not in locals():\n print(\"\\nSearching for 'Grains, total':\")\n for i in
  ↪  range(header_row + 1, len(df_65plus)):\n   food_group = df_65plus.iloc[i, 0]\n   if 'Grains, total' in
  ↪  str(food_group):\n     print(f\"Row {i}: {food_group} - {df_65plus.iloc[i, 1]}\")"}
 }
}
</tool_call>

user

<tool_response>
{"code": 0,
```

```
 "stdout": "Header row at index: 2\n2003-2004 found at column index 9\nMean column: 9, SE column:
↪  10\n\nSearching for 'Grains, total':\nRow 157: Grains, total - Total\nRow 158: Grains, total - FAH\nRow
↪  159: Grains, total - FAFH\nRow 160: Grains, total - FAFH: Restaurant\nRow 161: Grains, total - FAFH:
↪  Fast food\nRow 162: Grains, total - FAFH: School\nRow 163: Grains, total - FAFH: Others",
 "stderr": ""}
</tool_response>

assistant

<think>
We are looking for 'Grains, total' and 'FAFH: Restaurant' in the 'Ages 65 and above' sheet.
From the output, we see that row 160 has:
Food group: 'Grains, total'
Food source: 'FAFH: Restaurant'.
Let's get the value for 2003-2004 Mean from that row.
</think>

<tool_call>
{"name": "ExecuteCode",
 "arguments": {
  "code": "import pandas as pd\n\n# Load the Excel file\nfile_path = 'food-consumption-nutrient.xlsx'\n\n#
↪  Load the 'Ages 65 and above' sheet\ndf_65plus = pd.read_excel(file_path, sheet_name='Ages 65 and
↪  above', header=None)\n\n# Find the header row\nheader_row = None\nfor i in range(len(df_65plus)):\n if
↪  'Food group' in str(df_65plus.iloc[i, 0]):\n  header_row = i\n  break\n\nprint(f\"Header row at index:
↪  {header_row}\")\n\n# Find the column index for 2003-2004 Mean\nfor idx, val in
↪  enumerate(df_65plus.iloc[1]):\n if val == '2003{2004':\n  print(f\"2003-2004 found at column index
↪  {idx}\")\n  mean_col = idx\n  se_col = idx + 1\n  print(f\"Mean column: {mean_col}, SE column:
↪  {se_col}\")\n  break\n\n# Get the value for Grains, total from FAFH: Restaurant at row 160\nfood_group
↪  = df_65plus.iloc[160, 0]\nfood_source = df_65plus.iloc[160, 1]\nvalue = df_65plus.iloc[160,
↪  mean_col]\nprint(f\"\\nRow 160: Food group = '{food_group}', Food source =
↪  '{food_source}'\")\nprint(f\"Value for 2003-2004 Mean: {value}\")\n\n# Let's also print the entire row
↪  to verify\nprint(f\"\\nFull row 160:\")\nprint(df_65plus.iloc[160].tolist())"
 }
}
</tool_call>

user

<tool_response>
{"code": 0,
 "stdout": "Header row at index: 2\n2003-2004 found at column index 9\nMean column: 9, SE column: 10\n\nRow
↪  160: Food group = 'Grains, total', Food source = 'FAFH: Restaurant'\nValue for 2003-2004 Mean:
↪  3.06\nFull row 160:\n['Grains, total ', 'FAFH: Restaurant', 'Ounces per 1,000 calories', 2.71, 0.23,
↪  3.19, 0.15, 2.92, 0.14, 3.06, 0.2, 3.22, 0.25, 2.92, 0.13, 3.07, 0.17, 2.56, 0.22, 2.83, 0.19, 3.25,
↪  0.29, 3.4, 0.41]",
 "stderr": ""}
</tool_response>

assistant

<think>
First, I need to compute the expression:
(Energy from FAFH: Restaurant for Non-Hispanic Black)
minus
(Discretionary fats from FAFH: Restaurant for Hispanic)
minus
(Grains, total from FAFH: Restaurant for adults aged 65 and above).

From the data I extracted:
Energy from FAFH: Restaurant for Non-Hispanic Black: 138.7
Discretionary fats from FAFH: Restaurant for Hispanic: 21.45
Grains, total from FAFH: Restaurant for adults aged 65 and above: 3.06

Now, I need to perform the calculation: 138.7 - 21.45 - 3.06

Compute step by step:
138.7 - 21.45 = 117.25
117.25 - 3.06 = 114.19

So the resulting value is 114.19.

I should double-check the values to make sure I have the correct ones:
For Non-Hispanic Black: Energy from FAFH: Restaurant is 138.7
For Hispanic: Discretionary fats from FAFH: Restaurant is 21.45
For ages 65 and above: Grains, total from FAFH: Restaurant is 3.06

Yes, that seems correct.
</think>

<answer>
```

```
Based on the data from the Excel file, using 2003-2004 Mean values:

Energy from FAFH: Restaurant for Non-Hispanic Black population: 138.7
Discretionary fats from FAFH: Restaurant for Hispanic population: 21.45
Grains, total from FAFH: Restaurant for adults aged 65 and above: 3.06

The calculation is: 138.7 - 21.45 - 3.06 = 114.19

The resulting value is 114.19.
</answer>
```

