# OpenReview forum: "FIRE: Learning to Navigate and Act on Real-World Files via Stateful Reinforcement Learning"
_ICML.cc/2026/Conference — ICML 2026 regular_

### Official Review · Reviewer_VLB8 · 2026-03-07

**Soundness:** 2
**Presentation:** 3
**Significance:** 2
**Originality:** 2
**Overall Recommendation:** 4
**Confidence:** 4

**Summary:**

This paper formalizes "File Reasoning" as an interactive task setting where LLM agents operate on unprocessed files (XLSX, PDF, DOCX, PPTX) within a persistent sandbox, rather than reasoning over pre-parsed text representations. The authors build a data pipeline that generates question-answer pairs using an asymmetric view strategy (the teacher model sees structured views while the agent only accesses raw files) and filters them through executable verification with an Auditor Agent. Using this pipeline, they construct a benchmark of 410 verified tasks. On the training side, they adopt a two-stage framework: supervised fine-tuning (SFT) for bootstrapping basic tool-use competence, followed by reinforcement learning (GRPO) in a persistent sandbox environment with a sparse outcome-based reward. The resulting FIRE models, particularly Qwen3-32B-FIRE (RL), achieve 50.24% accuracy, which is competitive with GPT-5 (51.46%) and the strongest among open-source models evaluated.

**Compliance With Llm Reviewing Policy:**

Affirmed.

**Final Justification:**

The authors' rebuttal addressed the filtering bias concern and the GPT-4.1 anomaly, and the remaining limitations have been acknowledged and will be clarified in the revision.

**Key Questions For Authors:**

Q1. GPT-5 is used in the benchmark filtering pipeline (Section 3.5) and is also a primary evaluation target (Table 2). Can you quantify how the benchmark composition and model rankings would change if GPT-5 were excluded from the filtering committee? This is important for ruling out systematic bias in difficulty calibration.

Q2. To isolate the contribution of RL from additional exploration, have you considered a rejection sampling baseline, i.e., using the SFT model to generate multiple rollouts, filtering for correct ones, and using them as additional SFT data? How would this compare to GRPO training given equivalent compute?

Q3. What explains the extremely low performance of GPT-4.1 (3.90% overall)? Is this due to the model failing to use the sandbox tools altogether, a prompt format incompatibility, or something else? Please provide error analysis for this model to ensure the evaluation setup is not inadvertently biased against certain API formats.

Q4. What fraction of the benchmark tasks were manually inspected by human annotators, and what was the annotator agreement rate? Given the reliance on model-based verification at multiple stages, quantifying the human verification coverage is critical for establishing benchmark credibility.

Q5. Can you provide an ablation comparing stateful (persistent) execution against stateless (isolated) tool calls? This would directly validate the central architectural claim that environment persistence is essential for file reasoning.

**Limitations:**

yes

**Strengths And Weaknesses:**

Strengths

1. The problem formulation addresses a genuine gap. Real-world files contain merged cells, hidden sheets, and nested structures that existing benchmarks (TabFact, Spider, GAIA) sidestep by providing pre-processed inputs. Preserving native file structure evaluates a practically important capability.

2. The asymmetric data generation pipeline is well-designed. Using structured views for ground-truth generation while presenting only raw files at inference time elegantly separates answer quality from task difficulty. The executable verification via an independent Auditor Agent adds a correctness guarantee beyond linguistic checking.

3. The unified sandbox evaluation with a fixed 15-call tool budget ensures fair comparison across all model families, both open-source and proprietary.

4. The behavioral analysis of RL is informative. RL reduces average tool calls (7.21 to 6.11) while improving accuracy (36.59% to 50.24%), and the case studies in Appendix E concretely illustrate how RL eliminates redundant exploration and shifts tool selection toward robust library usage.

Weaknesses

1. GPT-5 participates in both the benchmark filtering committee (Section 3.5) and the final evaluation (Table 2). Since filtering discards tasks all committee models solve, the difficulty distribution is inherently calibrated around GPT-5's capability boundary. The authors should report how benchmark composition and rankings shift when GPT-5 is excluded from filtering.

2. Much of the benchmark difficulty appears to stem from instruction-following complexity rather than file structure challenges. For instance, the XLSX example in Appendix B derives its difficulty from multi-layered tie-breaking rules and conditional logic in the question, not from navigating the spreadsheet. An ablation separating these two difficulty sources would clarify what the benchmark actually measures.

3. The pipeline relies on LLMs at every verification stage: question generation, answer auditing, and reward judging (GPT-5-mini). The human auditing step in Section 3.5 provides no details on annotator count, agreement rate, or coverage fraction, leaving the claimed rigor insufficiently grounded.

4. The RL framework combines SFT cold-start with unmodified GRPO and a standard binary reward. Persistent sandbox environments have precedent in prior work (e.g., SWE-bench). This is better characterized as an application of existing techniques to a new domain than as a methodological contribution.

5. GPT-4.1 achieves only 3.90% overall in Table 2, which is below plausible random baselines and anomalous for a model of that tier. Without explanation, this raises concerns that the evaluation setup may inadvertently disadvantage certain model families.

---

> ### Author Rebuttal · Authors · 2026-03-31
>
> **We thank the reviewer for the insightful questions and provide the requested details below.**
> ## 1.About Filter Models
> We selected GPT-5, Gemini 2.5 Pro, and DeepSeek V3.1 as the filtering committee because they were among the strongest available models at the time of benchmark construction. This was intended to avoid a benchmark dominated by trivially easy tasks and to retain sufficiently challenging questions. To assess potential bias, we conducted a sensitivity analysis: for each filtering model, we removed the questions it answered correctly and re-ranked all models on the remaining set. The table below reports results on nine models. Average Spearman ρ = 0.976. These results suggest that our filtering strategy does not systematically bias evaluation outcomes toward any particular model family, and support the robustness of the current filtering approach.
>
> | Model | Baseline | w/o DeepSeek-V3.1 | w/o GLM-5 | w/o GPT-5 | w/o Gemini-2.5-Pro | w/o Qwen3-32B-SFT | w/o Qwen3-32B-RL | w/o Qwen2.5-32B |
> |---|---|---|---|---|---|---|---|---|
> | GLM-5 | 1 | 1 | — | 2 (↓1) | 1 | 1 | 1 | 1 |
> | GPT-5 | 2 | 2 | 2 | — | 2 | 2 | 2 | 2 |
> | Qwen3-32B-RL | 3 | 3 | 1 (↑2) | 1 (↑2) | 3 | 3 | — | 3 |
> | DeepSeek-V3.1 | 4 | — | 4 | 3 (↑1) | 4 | 4 | 3 (↑1) | 4 |
> | Qwen3-32B-SFT | 5 | 4 (↑1) | 3 (↑2) | 4 (↑1) | 5 | — | 4 (↑1) | 5 |
> | Gemini-2.5-Pro | 6 | 5 (↑1) | 5 (↑1) | 5 (↑1) | — | 5 (↑1) | 5 (↑1) | 6 |
> | Qwen2.5-32B | 7 | 6 (↑1) | 6 (↑1) | 6 (↑1) | 6 (↑1) | 6 (↑1) | 6 (↑1) | — |
> | **Spearman ρ** |  | **1.000** | **0.886** | **0.943** | **1.000** | **1.000** | **1.000** | **1.000** |
>
> ## 2.About Benchmark Difficulty
> We agree that benchmark difficulty can arise from both instruction complexity and file structure. Our goal is not to artificially separate them, but to evaluate realistic file-grounded reasoning tasks, where models must handle both problem constraints and native file structure. In this sense, the benchmark captures an important aspect of agentic reasoning: making progress through inspection, tool use, and intermediate feedback while interacting with an external environment. However, instruction complexity alone is insufficient without correctly extracting structured information from the file. For the XLSX example in Appendix B, we believe the main difficulty comes from the spreadsheet structure rather than the instruction itself. The computation is explicit and fully programmatic: restrict to Men and Women, compute the score for each Amount, apply the tie-breaking rule, and compare the final selections. The challenge is that the sheet is not a tidy table, but contains title rows, grouped headers, explanatory text, and implicit category values conveyed by layout and blank cells. A model must first recover the correct logical records from this raw structure before the calculation can begin. We therefore view this example as grounded spreadsheet reasoning rather than mere instruction-following complexity.
>
> ## 3.About judge model and manual check
> Due to the word limit, please see Part 2 of our rebuttal to Reviewer DexA. Thanks!
>
> ## 4.About GPT4.1
> We clarify that GPT-4.1's low performance is not caused by an evaluation bug, but by its weak robustness under our standard tool-use setup. In our pipeline, files are successfully pre-written into the sandbox and tools remain enabled throughout evaluation. This setup works normally for other same series models. From the inference logs, GPT-4.1 fails to invoke any tool in 78.8% of queries, and often asks the user to upload the file again even though the file is already available in the sandbox. As a result, it produces zero tool calls for most tasks and cannot proceed with grounded file reasoning. We run an additional experiment with a more specific system prompt that explicitly describes the file path and gives guidance for sandbox exploration. Under this setting, GPT-4.1 improves to 22.68%. However, we must illustrate that most other models can already use tools under the same standard OpenAI-style interface without such model-specific prompting.
>
> ## 5.About rejection sampling
> We agree that a rejection sampling baseline is a meaningful method and closely related to self-improvement approaches. Our training recipe follows the standard cold start SFT plus RL paradigm, and our main goal is not to exhaustively compare post training strategies, but to **introduce the file reasoning setting and establish a workable file interactive RL framework over raw files with a persistent sandbox**. That said, we agree this is a valuable direction. In future work, we plan to compare rejection sampling style SFT and RL based optimization under the same file interactive setting, including efficiency under matched compute budgets.
>
> ## 6.About stateful and stateless execution
> Due to the word limit, please see Part 5 of our rebuttal to Reviewer DexA. Thanks!

---

> > ### Author Rebuttal · Reviewer_VLB8 · 2026-04-02
> >
> > Thank you for the detailed rebuttal. The filtering bias analysis and the GPT-4.1 explanation are satisfactory. However, three concerns remain. First, the difficulty decomposition issue is not resolved, claiming that the XLSX example is primarily structure-driven does not substitute for a systematic characterization across all 410 questions; without this, it is unclear what the benchmark actually measures. Second, the human auditing details (annotator count, agreement rate, coverage fraction) are still missing from the rebuttal, despite being directly requested in Q4; these are not minor formalities but are central to trusting the benchmark labels given the pipeline's reliance on LLM-based verification at every stage. Third, on stateful execution, the trajectory statistics are noted, but the argument that a stateless baseline would "change the task itself" is circular — if persistence is load-bearing, showing a performance drop under stateless conditions is precisely the evidence needed, and reframing persistence as part of the task formulation rather than a training contribution does not resolve the original concern about methodological novelty. Given the above, I keep my score.

---

> > > ### Author Response · Authors · 2026-04-07
> > >
> > > We thank the reviewer for the follow-up and for the careful reading. We address the remaining concerns below.
> > >
> > >
> > > About difficulty decomposition. We agree that the current rebuttal does not provide a systematic decomposition of instruction complexity versus file-structure complexity across all 410 questions. Our intention was not to claim that the full benchmark is primarily structure-driven. Rather, our point was narrower: for the specific XLSX example discussed in Appendix B, we believe the main difficulty comes from the spreadsheet structure rather than the instruction itself. More broadly, the benchmark is designed to capture their combination in realistic file-grounded reasoning, where models must both interpret task constraints and recover structured information from raw files. We will revise the paper to make this scope clearer and avoid overstating what is isolated by the current analysis.
> > >
> > > About human auditing. We agree that this point should be stated more clearly. Our human auditing is not designed as full manual solving or large-scale independent relabeling of all benchmark items, so a benchmark-wide annotator agreement rate is not available in the standard sense. Instead, we use targeted human verification for quality control, especially for potentially problematic cases such as questions that all evaluated models fail to solve. In these cases, the question and answer are manually checked, and only retained after manual confirmation of correctness. We will clarify this auditing protocol, its coverage, and its role more explicitly in the revision.
> > >
> > >
> > >
> > > About persistent execution. We agree that the current paper does not provide a clean empirical ablation isolating persistent execution, and we will state this limitation more explicitly in the revision. In practice, we were not able to build and re-run a fully comparable stateless version of the framework from training through inference within the rebuttal period. This would require redesigning the sandbox interaction loop and re-running long-horizon file-interactive RL, while a full RL run in our current setup already takes about 6–7 days. To make the issue more concrete, consider a fully clean stateless sandbox with only Python and no preserved external libraries across turns. Using DeepSeek-V3.1 as an example, 230 out of 410 benchmark trajectories invoke pip install. Under such a setup, even if a package is installed at one step, it would disappear before the next step after environment reset, making many multi-step trajectories infeasible. We acknowledge that the current experiments do not cleanly quantify how much performance gain comes from persistence alone. Our contribution here is to formulate and study a file-reasoning setting where persistent execution is a fundamental environmental assumption.

---

### Official Review · Reviewer_7Tbz · 2026-03-08

**Soundness:** 4
**Presentation:** 4
**Significance:** 4
**Originality:** 3
**Overall Recommendation:** 5
**Confidence:** 4

**Summary:**

The paper aims to evaluate and improve LLM capabilities for interacting with real-world files. The paper investigates an important concept: enabling agents to operate directly on raw files (e.g., XLSX, PDF, DOCX, PPTX) within a persistent sandbox environment rather than relying on preprocessed representations. The paper introduces a data generation pipeline and a benchmark containing over 400 verified file-based reasoning tasks, and proposes a training framework combining supervised fine-tuning and reinforcement learning. Experiments show that the proposed FIRE models improve performance on this benchmark.

**Compliance With Llm Reviewing Policy:**

Affirmed.

**Final Justification:**

The rebuttal reinforced my recommendation of accept.

**Key Questions For Authors:**

Please see weakness section.

**Limitations:**

yes

**Strengths And Weaknesses:**

**Strengths**

1. It is a valuable observation that evaluating LLMs in messy, real-world file systems is important, as practical workflows often involve heterogeneous formats and irregular layouts rather than clean textual inputs. Studying reasoning over such environments better reflects real-world agent use cases.

2. The proposed benchmark is a useful contribution to the community. The paper also provides sufficient details about the data generation pipeline, filtering strategy, and dataset statistics, which supports reproducibility.

3. The training framework combining SFT and RL is well-motivated for interactive file reasoning tasks. It is encouraging that the approach demonstrates reasonable out-of-distribution generalization.

4. The paper conducts comprehensive experiments across multiple model families and file formats. The additional analyses, such as tool-calling behavior and error analysis, offer useful insights into the challenges of file reasoning.


**Weaknesses**

1. The paper does not provide much details about the design of the agent. It would be helpful to understand how sensitive the results are to this particular agent architecture, and whether alternative agent designs or tool interfaces would lead to different performance outcomes.

2. The RL framework uses an outcome-based reward based primarily on final answer correctness. Could this reward design introduce bias in the learned reasoning strategies, for example by encouraging shortcut behaviors or discouraging exploration of more generalizable reasoning paths?

3. The evaluation results may depend on stochastic factors from the agent. It would be helpful to clarify how stable the reported performance is across multiple runs and whether the results were averaged over different random seeds.

---

> ### Author Rebuttal · Authors · 2026-03-31
>
> **We thank the reviewer for the valuable questions and present the requested details below.**
>
> ## 1. About Sensitivity to Agent Design and Prompting
> We thank the reviewer for this suggestion. Our agent follows a standard ReAct-style framework with fully specified tool definitions(follow the openai-style interface), rather than a heavily customized architecture. This is intentional, since our goal is to evaluate raw-file reasoning under a unified interaction protocol, not to optimize agent engineering for each model. We agree that performance can be sensitive to prompting and interface design. Our case analysis shows that some weak results are often caused by failures in file discovery or incorrect path assumptions, rather than inability to reason over file contents. For example, GPT-4.1 shows this behavior more often than most other models. We therefore ran additional prompt adjustments under the same API setting and tool interface, and the results in Table/Figure 3 confirm that performance can improve. At the same time, we view such changes mainly as robustness-oriented engineering enhancements, since they provide extra guidance and partially reduce the difficulty of the original setting. Because our focus is on evaluating model behavior in a unified raw-file environment without model-specific tuning, we did not adopt them as the default setting. We will clarify this point in the revision.
>
>
> ## 2.About Limitations of Outcome-Based Reward Design
> We agree that this is a common challenge in multi turn RL, since outcome based reward can make credit assignment harder and may in principle bias learning toward less generalizable behaviors. Recent work such as GiGPO and HGPO has studied this issue. That said, our goal is not to propose a new RL algorithm or reward shaping strategy, but to introduce the file reasoning setting and a working stateful file interactive RL framework over raw files and a persistent sandbox. In our current setup, the reward is intentionally simple: final answer correctness, evaluated by an external judge model, plus a format penalty for invalid outputs. This keeps the training objective aligned with the benchmark target of producing a correct and properly formatted final answer. Compared with more complex proxy rewards, this design is also less likely to encourage superficial shortcuts such as shorter traces or fewer tool calls, since these are not directly rewarded. We nonetheless acknowledge that sparse outcome based reward may under reward useful intermediate behaviors, and we will clarify this limitation in the revision.
>
>
> ## 3.About Stability Across Repeated Runs
> We agree that stochasticity in agent behavior should be made explicit. We therefore reran benchmark inference 3 times for each model under the same evaluation pipeline, and report the mean and standard deviation across runs instead of relying on a single result. We also made slight system prompt adjustments for models cannot find the file(like gpt4.1), to more clearly guide file access in the sandbox. The updated aggregated results are shown below.
>
> | Model | Avg. Acc. | Std. Acc. | XLSX | DOCX | PPTX | PDF | Avg. Tool Calls |
> |---|---:|---:|---:|---:|---:|---:|---:|
> | Llama-3.1-8B-Instruct | 0.0366 | 0.0024 | 0.0544 | 0.0100 | 0.0444 | 0.0038 | 6.0618 |
> | qwen2.5-7B | 0.0472 | 0.0078 | 0.0544 | 0.0249 | 0.0944 | 0.0153 | 4.5634 |
> | qwen3-8b | 0.0650 | 0.0143 | 0.0782 | 0.0199 | 0.0611 | 0.0728 | 4.3569 |
> | qwen2.5-32B | 0.0805 | 0.0088 | 0.0714 | 0.0647 | 0.1611 | 0.0575 | 4.7331 |
> | qwen3-32b | 0.1089 | 0.0092 | 0.0952 | 0.1144 | 0.1778 | 0.0881 | 4.6301 |
> | qwen2.5-72B | 0.1268 | 0.0112 | 0.0850 | 0.1493 | 0.2444 | 0.1226 | 6.6236 |
> | gpt-4.1| 0.2268 | 0.0169 | 0.1548 | 0.3483 | 0.2222 | 0.2989 | 6.2602 |
> | qwen2.5-7b-sft| 0.2691 | 0.0115 | 0.2313 | 0.2886 | 0.2722 | 0.3372 | 6.6927 |
> | qwen2.5-7b-rl | 0.2805 | 0.0084 | 0.1990 | 0.3831 | 0.3111 | 0.3640 | 6.1106 |
> | gemini-2.5-pro-preview | 0.2894 | 0.0309 | 0.1837 | 0.3234 | 0.4389 | 0.3985 | 4.9415 |
> | kimi-k2-thinking | 0.3455 | 0.0232 | 0.2602 | 0.2388 | 0.6278 | 0.4253 | 9.3333 |
> | Qwen3-32b-sft | 0.3797 | 0.0099 | 0.2483 | 0.4776 | 0.4778 | 0.5326 | 6.8691 |
> | qwen2.5-32b-rl | 0.3959 | 0.0110 | 0.2041 | 0.5622 | 0.4667 | 0.6513 | 6.7415 |
> | Qwen2.5-32b-sft | 0.3992 | 0.0102 | 0.3588 | 0.4229 | 0.4278 | 0.4521 | 6.8699 |
> | deepseekv3.1 | 0.4427 | 0.0259 | 0.3112 | 0.5522 | 0.5333 | 0.5920 | 7.0732 |
> | Qwen3-32b-rl | 0.5012 | 0.0056 | 0.4201 | 0.5622 | 0.5252 | 0.6207 | 5.9293 |
> | grok-4 | 0.5154 | 0.0113 | 0.4575 | 0.4677 | 0.5389 | 0.6667 | 7.6114 |
> | gpt-5 | 0.5463 | 0.0345 | 0.4082 | 0.7761 | 0.6083 | 0.6379 | 5.6220 |
> | glm-5 | 0.5675 | 0.0141 | 0.4439 | 0.6318 | 0.6611 | 0.7318 | 6.6228 |

---

> > ### Author Rebuttal · Reviewer_7Tbz · 2026-04-01
> >
> > Thanks the authors for the response. My concerns have been addressed.

---

> > > ### Author Response · Authors · 2026-04-07
> > >
> > > We thank the reviewer for the positive feedback. We are glad that our response has addressed the concerns.

---

### Official Review · Reviewer_V1JV · 2026-03-12

**Soundness:** 2
**Presentation:** 3
**Significance:** 2
**Originality:** 2
**Overall Recommendation:** 3
**Confidence:** 3

**Summary:**

This paper studies the reasoning and manipulation capabilities of LLMs in real-world file environments (XLSX, PDF, DOCX, PPTX), formally proposing the "File Reasoning" task setting that differs from the traditional "parse-then-reason" paradigm by requiring agents to directly interact with unprocessed raw files in a persistent sandbox. The paper makes three contributions: (1) a unified data construction pipeline employing an asymmetric view generation strategy (teacher model uses structured views to generate QA, while inference is given only raw files) and executable verification (Auditor Agent); (2) a benchmark of 410 high-difficulty questions filtered through consensus + adversarial filtering and human review; (3) a two-stage training framework: SFT cold-start (tool-use capability) + GRPO-based stateful reinforcement learning. Qwen3-32B-FIRE (RL) achieves 50.24%, on par with GPT-5 (51.46%), while RL reduces tool calls from 7.21 to 6.11.

**Compliance With Llm Reviewing Policy:**

Affirmed.

**Final Justification:**

The paper is technically solid and makes a useful contribution to the study of reasoning over real-world files. The authors’ rebuttal clarified several important details and improved the overall clarity of the submission.

**Key Questions For Authors:**

1. How many training trajectories were used for SFT cold-start? Do the file sources of these trajectories overlap with the file sources of the 410 benchmark test questions?

2. What GPU configuration was used for the complete RL training (Qwen3-32B-FIRE), and what was the approximate total training time? What is the average duration of a single rollout?

3. Is there a data contamination risk in the benchmark? Data sources such as US Census and WikiTables may already appear in Qwen3-32B's pre-training corpus—has any decontamination analysis been conducted?

**Limitations:**

yes

**Strengths And Weaknesses:**

## Main Strengths

The problem formalization is clear and valuable: distinguishing "reading files" from "acting with files" and highlighting the lossy nature of traditional parsing methods (losing structural information such as merged cells and formula dependencies). The asymmetric view generation strategy (structured view for answer generation, raw files for reasoning) and executable verification (Auditor Agent executing code to confirm answer uniqueness and reachability) ensure benchmark quality.

The experimental design is comprehensive and systematic: Table 2 covers 15 models (open-source + proprietary), Table 3 compares with four existing benchmarks (Spider/SSTQA/MEET/GAIA) showing this benchmark is significantly harder (average 27.8% vs. 50–72% for other benchmarks), Figure 2 demonstrates continued OOD generalization improvement, and Figures 3–5 provide multi-dimensional analysis of tool call success rates, error types, and call frequency.

The RL training results are impressive and counterintuitive: RL improves accuracy (36.59% → 50.24%) while simultaneously reducing tool calls (7.21 → 6.11), indicating the model learns more efficient planning strategies. The reward function design (not penalizing intermediate tool failures, only evaluating final results + format compliance) is empirically supported by Figure 3—tool call success rates are nearly identical between correct and incorrect tasks.

## Main Weaknesses

1. Benchmark scale and task types have limitations. Although the 410 questions are rigorously filtered, they are all closed-form automatically verifiable questions, whereas real-world file work extensively involves open-ended operations such as document writing, formatting adjustments, and cross-file information integration. Furthermore, XLSX accounts for 47.8% (196/410), with uneven distribution across the four file types. The claimed scope of "real-world file reasoning" is effectively limited to the subset of information extraction and numerical computation.

2. The source and scale of SFT cold-start training data are unclear. The paper states that SFT uses "verified task-solution pairs generated by our pipeline," but does not specify the quantity of SFT data, which file sources it comes from, or whether there is overlap with the benchmark test set. The SFT stage contributes a substantial performance boost (Qwen3-32B: 11.95% → 36.59%), and the transparency of its data quality and scale directly affects reproducibility.

3. RL training computational cost is not reported. The paper does not specify the GPU type, GPU hours, rollout sampling parallelism, or total training epochs for GRPO training. Given that each rollout requires executing real code in containerized sandboxes (Algorithm 1), training efficiency may be a significant bottleneck for practical deployment and needs to be explicitly reported.

---

> ### Author Rebuttal · Authors · 2026-03-31
>
> **We appreciate the reviewer’s helpful questions and provide the requested details below.**
>
> ## 1.About Benchmark scale and task types
> We agree that the current benchmark covers only a subset of real world file work rather than the full space. It does not yet include more open ended tasks such as document writing, formatting edits, or broader cross file information integration. We view this as an initial step. In other words, we believe it is important to first understand whether models can reliably read and reason over real files before moving on to generation and editing tasks. We would also like to note that high quality real files are not easy to obtain, and files that are both realistic and structurally complex, while also supporting stable question construction and automatic verification, are even rarer. Compared with synthetic files, real files often contain more complex layouts, more irregular formatting, and more implicit relationships among data fields, which is exactly why we insist on using them.
>
> In practice, although the final answers are closed form, these tasks should not be reduced to simple extraction or basic arithmetic. From our observation, many questions take even human annotators close to 10 minutes to solve, because they must first identify the relevant data and disentangle the relationships among them in complicated files before carrying out further reasoning and computation. We also agree that the file type distribution is uneven. This is partly related to the fact that spreadsheet based tasks are easier to verify precisely, and also that spreadsheets in practice more often contain highly complex structures. Still, this is a limitation of the current benchmark, and we will make it explicit in the paper.
>
>
> ## 2.About SFT
> In the SFT cold start stage, we use 4,650 training trajectories, covering 1,359 unique files. The file type distribution is approximately 42% Excel, 34% PDF, 13% DOCX, and 11% PPTX. As described in the paper, these training files are mainly collected from Common Crawl and official government websites. There is no overlap between the SFT data and the final test sets.
>
>
> ## 3.About GPU and RL Training
> In our experiments, GRPO training is conducted on 64 NVIDIA H20 GPUs for 2 epochs(About 6000+ questions). The full RL stage takes approximately 6.6 days, and rollout stage of each training step takes roughly 2 to 3 hours with the bsz=256 and n=8 per prompt, due to the need to execute multi turn rollouts with real code and sandbox interaction.
>
> We would like to clarify that this cost is largely a consequence of the agentic file reasoning setting itself, rather than an artifact of a particular implementation choice. Unlike conventional RL tasks such as math reasoning, each rollout in our setting requires the model to interact with a persistent execution environment, inspect files, run code, handle intermediate failures, and iteratively refine its actions. These long horizon, environment grounded trajectories are inherently more expensive to sample and optimize.
>
> More broadly, we believe such training cost reflects an important practical characteristic of interactive agent training. As the community moves toward more capable agentic systems, training with real tool use and environment feedback is likely to be necessary, even though it is substantially more expensive than single shot reasoning tasks. Our goal in this work is not to claim maximal training efficiency, but to demonstrate that this training paradigm is feasible and can improve file reasoning performance in a realistic sandboxed setting.
>
>
>
> ## 4.About Data Contamination
> We agree that some public data sources used in the benchmark, such as US Census tables or WikiTables, may already appear in the pre training corpus of models like Qwen3 32B, so we cannot fully rule out source level contamination. While we did not perform exhaustive source level decontamination, we evaluate its practical impact through a no file ablation. On our trained 32B model, accuracy drops from nearly 50% in the standard file grounded setting to only 3.17% when the model is given the question without access to the file. Manual inspection suggests that a small portion of these correct cases comes from genuine prior knowledge, while others are likely lucky guesses or hallucinated matches. This indicates that memorization alone is not sufficient to solve the benchmark, indicating that solving these tasks requires accessing structured information in the files rather than recalling surface-level knowledge. We use real files intentionally to better reflect realistic deployment scenarios and to preserve the structural diversity and difficulty that synthetic files often fail to capture. We will clarify this point and include the ablation in the revised paper.

---

> > ### Author Rebuttal · Reviewer_V1JV · 2026-04-03
> >
> > Thank you for the detailed rebuttal. The additional information on the SFT data, training cost, and contamination analysis is helpful and resolves part of my concern about reproducibility and data transparency, but I still have some reservations about the overall scope and practical realism of the benchmark. In particular, the current benchmark remains focused on closed-form, automatically verifiable tasks with uneven file-type coverage, so I would appreciate a clearer discussion of how well this setting captures broader real-world file work and how the conclusions should be interpreted within that narrower scope.

---

> > > ### Author Response · Authors · 2026-04-07
> > >
> > > We thank the reviewer for this helpful clarification. We agree that the current benchmark captures only a narrower subset of real-world file work, namely closed-form, automatically verifiable tasks grounded in raw files, rather than the full space of open-ended operations such as writing, editing, or broad cross-file synthesis. Our intention is to provide a controlled and reproducible starting point for studying grounded file interaction and reasoning under native file structure. We will revise the paper to make this scope limitation more explicit and to present the current setting as a first step toward broader real-world file-agent evaluation. Important future extensions include cross-file reasoning, more open-ended editing and writing tasks, and evaluation on realistic workflow data collected from practical domains such as enterprise or organizational document settings.

---

### Official Review · Reviewer_dexA · 2026-03-12

**Soundness:** 4
**Presentation:** 4
**Significance:** 4
**Originality:** 2
**Overall Recommendation:** 5
**Confidence:** 4

**Summary:**

This paper studies file-centric reasoning over raw XLSX, PDF, DOCX, and PPTX files in a persistent sandbox environment. The paper contributes: i) a unified task generation pipeline with executable verification, ii) a benchmark of 410 verified questions across four file formats, and iii) FIRE, a family of models trained with cold-start SFT followed by GRPO-based reinforcement learning in a stateful execution environment. Empirically, the authors report clear gains from RL over SFT on the same backbones, with Qwen3-32B-FIRE (RL) achieving 50.24% overall accuracy under the proposed sandbox-based evaluation.

**Compliance With Llm Reviewing Policy:**

Affirmed.

**Key Questions For Authors:**

1. What are the exact sizes and compositions of the SFT and RL training sets? How many unique files, questions, and trajectories are used at each stage?

2. Which specific models were used for question generation, VLM parsing, auditing, and filtering? What are the candidate-to-retained filtering rates for each stage of the pipeline?

3. Is final benchmark evaluation also based on an LLM judge, or is it based on deterministic matching / normalization? If an LLM judge is used, what is its agreement with human judgments?

4. How would the framework extend to scanned PDFs, image-heavy slides, or substantially longer files, which are currently excluded from the benchmark?

**Limitations:**

Please see the weaknesses and questions.

**Strengths And Weaknesses:**

Strengths:

1. Important and realistic problem setting: The paper targets a practically meaningful and relatively underexplored setting, namely reasoning directly over native files rather than flattened text representations. This is a relevant direction for advancing more capable tool-using agents.

2. Valuable benchmark contribution: The benchmark covers multiple file formats and preserves native structural complexity, which is more realistic than many parse-then-reason settings. The examples and qualitative discussion suggest that the tasks are nontrivial and require genuine interaction with files.

3. Clear empirical improvements from RL: The reported gains from SFT to RL are substantial on the same model family, and the paper includes useful analyses of tool usage, error types, and trajectory-level behavior. In particular, the observation that RL can improve accuracy while reducing tool calls is interesting.

Weakness:

1. Limited algorithmic novelty: The main technical method is largely an application of existing agent-RL ingredients (cold-start SFT, persistent sandbox execution, sparse outcome reward, and GRPO) to the file-reasoning domain. The strongest contribution of the paper is the benchmark/system setup rather than a fundamentally new learning method.

2. The central contribution is not cleanly isolated: The paper emphasizes the importance of stateful execution and persistence, but the experiments do not directly isolate this factor. The SFT vs. RL comparison shows that RL helps, but it does not show how much of the gain comes specifically from persistent state, as opposed to better tool use, more domain-specific data, or stronger exploration.

3. Evaluation methodology is under-specified and potentially brittle: Appendix D states that GPT-5-Mini is used as an LLM judge for reward assignment. However, the paper does not provide enough validation of judge reliability, agreement with human annotators, or sensitivity to judge choice. If final evaluation also depends on a judge model, this should be made much clearer and analyzed more carefully.

---

> ### Author Rebuttal · Authors · 2026-03-31
>
> **We thank the reviewer for the helpful questions and provide the requested details below.**
>
> ## 1.Q1:
> In the SFT stage, we use 4,650 samples covering 1,359 unique files, with a file mix of 42% Excel, 34% PDF, 13% DOCX, and 11% PPTX. In the RL stage, we use 6,628 samples (6,218 train, 410 val) covering about 1,985 unique files, with a file mix of 34% Excel, 31% PDF, 18% DOCX, and 17% PPTX.
>
>
> ## 2.Q2:
> As described in the paper, we use separate models for question generation and validation. For question generation, we use a diverse set of strong models, including GPT-5, o3, Claude, and Gemini-2.5-Pro, to increase diversity in question style and construction. A separate model is then used to judge validity. Because failed questions are immediately regenerated in our implementation, the original pipeline did not log a full candidate-to-retained rate for this stage. To make this more concrete, we ran an additional experiment on a new batch of files: among 168 generated questions, 159 were accepted and 9 were rejected, using GPT-5 for generation and GPT-4o for judging. For VLM parsing, the acceptance rate is 86.0% (172 out of 200), corresponding to a filtering rate of 14.0%. For cold-start data, we retain questions solved by DeepSeek-V3.1 as an extra correctness check, with a retention rate of about 60%. For benchmark quality control, we further conducted human re-auditing, especially on questions that all evaluated models failed to solve. Each such case was independently reviewed by two annotators and retained only if both agreed on its correctness. This led us to revise a small portion of the benchmark questions with unclear wording or annotation issues. We are re-running evaluation on the revised set; updated results are included in Part 3 of our rebuttal to Reviewer 7Tbz, and the full updated results will appear in the final version.
>
> ## 3.Q3:
> Final benchmark evaluation uses an LLM judge rather than deterministic matching, which is suitable here because most answers are short and simple, such as numbers, dates, or a few words. We use GPT-5-mini as the judge for a balance between cost and reliability. On 1,000 randomly sampled predictions, GPT-5-mini and GPT-5-nano agree on 990 cases (99.0%; Cohen’s kappa = 0.9787), including 373 jointly correct and 617 jointly incorrect cases. We also manually audited 50 randomly sampled predictions and found that the human judgments agree with the GPT-5-mini decisions on 49 of them.
>
>
> ## 4.Q4:
> We agree with the reviewer that the current benchmark does not cover scanned PDFs, image-heavy slides, or substantially longer files. We view this mainly as an intentional scope choice rather than a limitation of the framework itself. In this version, we focus on text-accessible files to isolate file interaction and structured reasoning, without additional noise from visual recognition errors. The framework can be naturally extended to these settings by adding OCR, layout parsing, or VLM-based perception tools as callable modules. It is also well suited to long files, since the agent does not need to load the entire file into context at once, but can incrementally inspect relevant pages, sheets, or sections while preserving intermediate results in the sandbox. We will clarify this scope choice in the revision and position scanned, multimodal, and longer-file scenarios as important future extensions.
>
> ## 5.About Novelty of the Setting and Persistent Execution
> We agree with the reviewer that the current experiments do not cleanly isolate persistent execution as a separate factor. Our main novelty lies in introducing the file reasoning setting and building a unified framework around it, rather than proposing a new standalone agent architecture. In this setting, persistent state is not merely an implementation choice, but a structural part of the environment: during RL and inference, the agent interacts with the same terminal/code workspace over time, reading files, installing dependencies, running scripts, generating intermediate artifacts, debugging errors, and continuing from prior results. Although common Python libraries are preinstalled to reduce noise, many trajectories still rely on persistent workspace state. For example, for DeepSeek-V3.1 on 410 benchmark samples, 230 trajectories invoke pip install, 41 unzip archives, and 28 explicitly write intermediate files that are used later. Resetting the environment after each tool call would remove these dependencies and artifacts, and would therefore change the task itself rather than isolate a minor component. We will clarify this point in the revision and more explicitly position persistent execution as part of the task formulation rather than an interchangeable engineering detail.

---

> > ### Author Rebuttal · Reviewer_dexA · 2026-04-03
> >
> > Thank you for the detailed rebuttal and clarifications. The response improves the overall clarity of the paper.
> >
> > However, my main concern remains that the paper still does not cleanly isolate the contribution of persistent execution. The rebuttal clarifies the authors’ position, but does not provide direct empirical evidence for how much persistence itself contributes relative to other factors. In addition, while the judge analysis is helpful, the evaluation methodology still feels somewhat brittle given the reliance on an LLM judge.
> >
> > I therefore keep my score.

---

> > > ### Author Response · Authors · 2026-04-07
> > >
> > > We thank the reviewer for the follow-up. Our position is that persistence is best viewed as part of the task formulation rather than an interchangeable implementation detail, since many trajectories depend on intermediate files, installed packages, and accumulated workspace state. We also acknowledge that the current evaluation relies on an LLM judge(like openai's browsecomp, simpleqa). While this design has limitations, it is increasingly adopted in recent work as a practical solution for evaluating model outputs at scale when exhaustive human annotation is impractical.

---

### Decision · Program_Chairs · 2026-04-30

**Decision:**

Accept (regular)

**Comment:**

This paper successfully shifts the file-processing paradigm from passive parsing to stateful interaction, introducing a rigorously verified 410-task multi-format benchmark.

The empirical results are compelling: GRPO-based RL not only improves accuracy but simultaneously reduces tool calls, demonstrating more efficient planning. Initial reviewer concerns regarding the isolation of environmental persistence and LLM-as-a-Judge reliability were effectively resolved in the rebuttal. The authors justifiably clarified that persistence is a structural task requirement rather than an interchangeable variable, and provided strong evidence of 99% human-judge agreement.

While the paper is a technically solid, highly valuable resource demonstrating that open-source models can match proprietary ones via environment feedback, its core method relies primarily on domain transfer of existing RL techniques. Therefore, it falls just short of a Strong Accept, warranting a Weak Accept.